# Highly variable iron content modulates iceberg-ocean fertilisation and potential carbon export

Mark J. Hopwood [1]*, Dustin Carroll [2], Juan Höfer [3,4], Eric P. Achterberg [1], Lorenz Meire [5,6], Frédéric A.C. Le Moigne[7], Lennart T. Bach [8], Charlotte Eich[9], David A. Sutherland [10] & Humberto E. González[4,11]

Marine phytoplankton growth at high latitudes is extensively limited by iron availability. Icebergs are a vector transporting the bioessential micronutrient iron into polar oceans. Therefore, increasing iceberg fluxes due to global warming have the potential to increase marine productivity and carbon export, creating a negative climate feedback. However, the magnitude of the iceberg iron flux, the subsequent fertilization effect and the resultant carbon export have not been quantified. Using a global analysis of iceberg samples, we reveal that iceberg iron concentrations vary over 6 orders of magnitude. Our results demonstrate that, whilst icebergs are the largest source of iron to the polar oceans, the heterogeneous iron distribution within ice moderates iron delivery to offshore waters and likely also affects the subsequent ocean iron enrichment. Future marine productivity may therefore be not only sensitive to increasing total iceberg fluxes, but also to changing iceberg properties, internal sediment distribution and melt dynamics.

[1] GEOMAR, Helmholtz Centre for Ocean Research Kiel, Kiel, Germany. [2] Moss Landing Marine Laboratories, San José State University, Moss Landing, CA, USA. [3] Escuela de Ciencias del Mar, Pontificia Universidad Católica de Valparaíso, Valparaíso, Chile. [4] Centro FONDAP de Investigación en Dinámica de Ecosistemas Marinos de Altas Latitudes (IDEAL), Universidad Austral de Chile, Valdivia, Chile. [5] Royal Netherlands Institute for Sea Research, and Utrecht University, Yerseke, The Netherlands. [6] Greenland Climate Research Centre, Greenland Institute of Natural Resources, Nuuk, Greenland. [7] Mediterranean Institute of Oceanography, UM110, CNRS, IRD, Aix Marseille Université Marseille, Marseille, France. [8] Institute for Marine and Antarctic Studies, University of Tasmania, Hobart, Tasmania, Australia. [9] Royal Netherlands Institute for Sea Research, and University of Amsterdam, Texel, The Netherlands. [10] Department of Earth Sciences, University of Oregon, Eugene, Oregon, USA. [11] Instituto de Ciencias Marinas y Limnológicas, Universidad Austral de Chile, Casilla 567, Valdivia, Chile. *email: mhopwood@geomar.de

cebergs have long been considered an important supply route of iron (Fe) to marine phytoplankton[1–3] and are hypothesized to be one of the largest Fe sources (3.9–30.5 Gmol Fe yr$^{-1}$)[4] to both the Arctic and Southern Oceans[5,6]. Increased biological activity following iceberg passage in the Southern Ocean is indicated by both satellite-derived chlorophyll[7–9] and limited in situ observations[10–12], supporting the hypothesis that icebergs are ocean Fe fertilizers. Ice discharge in both the Arctic and Antarctic has increased in response to recent climate change[13,14]; thus potentially increasing Fe supply to polar oceans, enhancing productivity and increasing the resultant carbon (C) export[7,11,15]. However, the magnitude and spatial distribution of iceberg Fe fertilization remains uncertain and has yet to be explicitly simulated in global ocean biogeochemical models[16], with regional models unable to achieve consensus on the significance of iceberg fertilization in the present-day ocean[17–20].

Direct observations of iceberg Fe concentrations and the magnitude of the resultant ocean fertilization are sparse, and to some extent contradictory. Duprat et al.[7], recently suggested, using satellite-derived chlorophyll data, that icebergs drive >20% of Southern Ocean particulate organic carbon export (POC). This is supported by one model estimate of 30% for Antarctic runoff and icebergs[17], yet other estimates of iceberg fertilization vary widely and are in some cases significantly smaller[8,9,18,19]. Similarly, there is disagreement about whether iceberg fertilization is more, or less, efficient as iceberg size increases[7,8]. Part of this uncertainty may arise due to the variable and poorly constrained mechanisms that distribute iceberg melt in surrounding waters and thus affect resource (light, macronutrient and micronutrient) availability in the water column[21,22]. Sporadic, localized upwelling driven by buoyant plumes of iceberg melt may result in surface Fe and macronutrient enrichment, but also phytoplankton dilution within a few kilometers of icebergs[9,12,23]. Conversely, iceberg basal and sidewall melt may form horizontal meltwater intrusions at depth that potentially enrich a larger

spatial area with Fe but have no surface biogeochemical signature close to icebergs[22,24].

A critical challenge in reconciling high and low estimates of iceberg fertilization is quantifying how much Fe is present in calved ice, and what fraction of this Fe enters the offshore, near-surface ocean. Estimates of this flux range from negligible, based on measured surface Fe distribution near some melting icebergs, to one of the largest Fe fluxes into the global ocean based on extrapolations from glacial sediments[25–27]. In order to reduce this uncertainty, we present a global dataset of Fe concentrations measured directly from 206 iceberg samples (Source Data File). Our data reveals that iceberg Fe content is highly variable with a total dissolvable Fe concentration range of 2 nM to 2 mM and regional median Fe concentrations normally in the range 44–790 nM. Combining this new dataset with numerical modeling, we constrain lower limits to the fraction of iceberg-derived Fe that can be transported to the open ocean demonstrating that Fe loss processes from icebergs are highly sensitive to the location of Fe-rich layers within ice. Using these results, we explore the implications for primary production and C export, highlighting the large uncertainties that remain concerning the fate of iceberg-derived Fe immediately after its discharge into the ocean and variability in marine C export efficiency.

## Results and Discussion

In order to constrain iceberg Fe content, iceberg samples were collected from the coastal periphery of Antarctica, Greenland, Svalbard, Iceland and Patagonia. Total dissolvable Fe (hereafter, Fe), Fe which is soluble in weak HCl after 6 months at pH <2, was determined from melted ice samples and represents the upper limit of potentially bioaccessible Fe in the marine environment[28]. Fe displays a 6-order of magnitude difference between the highest (1.9 mM) and lowest (2.1 nM) concentration (Fig. 1a, $n = 201$). Analysis of variance confirms that Fe in ice from Jökulsárlón (Iceland, range of 6.2 μM–1.9 mM) is significantly elevated

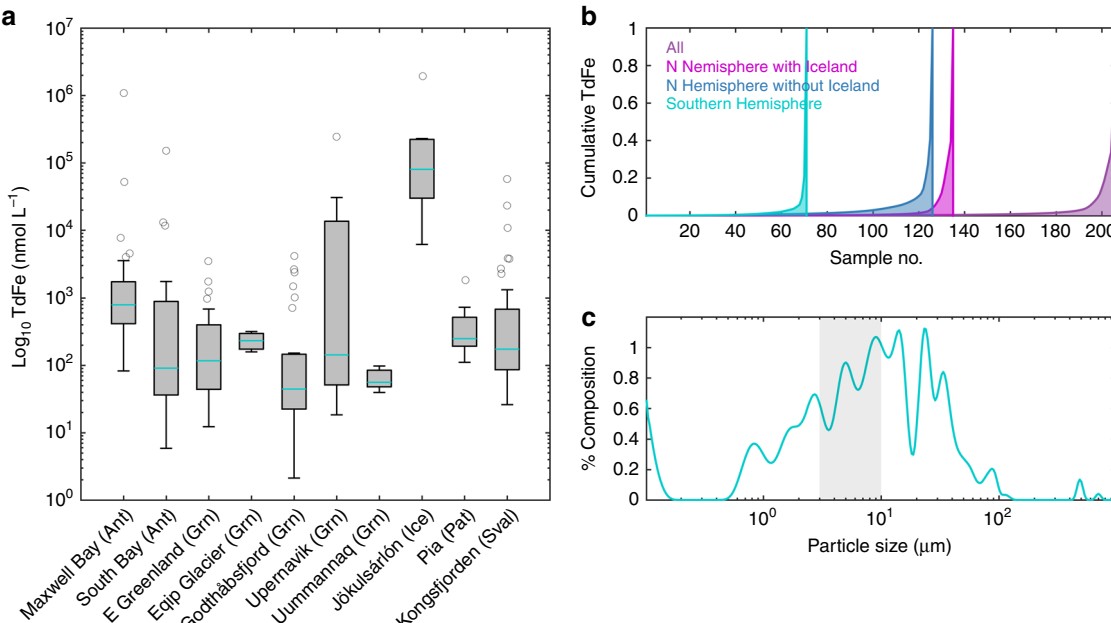

**Fig. 1** Iceberg iron and particle content. **a** Iceberg iron (total dissolvable Fe) content per catchment. Boxes show median, 25th and 75th percentiles; whiskers show 10th and 90th percentiles; dots mark all outliers. Regions: Antarctica (Ant), Greenland (Grn), Iceland (Ice), Patagonia (Pat), Svalbard (Sval). Data only shown for catchments where n > 4. Source data are provided as a Source Data File. **b** Cumulative distribution plots of the Fe dataset sorted by increasing Fe content. **c** Lithogenic iceberg-borne particle size. Percentage composition from analysis of sediment samples retained from icebergs in Svalbard ($n = 51$). Shaded area corresponds to coarse atmospheric dust

($p < 0.003$) relative to all other catchments. Differences in Fe content between all other catchments and regions were insignificant ($p > 0.9$, see Supplementary Note 2 and Supplementary Tables 1 and 2). When samples from Jökulsárlón are excluded, the median TdFe concentration for catchments ranges from 44–790 nM. This is consistent with the limited iceberg Fe measurements from previous studies (20–290 nM, $n = 5$)[3,29–31]. Furthermore, the global mean 9.3 µM Fe (excluding Jökulsárlón) is comparable to previous estimates of mean iceberg labile Fe content scaled-up from ascorbic acid leaches of glacial sediments and estimated iceberg sediment load (6.8 µM, using an estimate of 0.5 g sediment $L^{-1}$ in icebergs)[4,6]. The exceptionally high Fe (mean and median of 310 µM and 93 µM respectively) for Jökulsárlón is likely due to the volcanic ash that is visibly present within this ice. Whilst volcanic enrichment of glacier-ice is not unique to Iceland[32], it is uncommon and thus these data are excluded from the global mean for flux calculations.

With ice discharge from Greenland and Antarctica of ~500 and 1100 km$^3$ yr$^{-1}$[33,34], respectively, the mean (9.3 µM) global iceberg Fe concentration corresponds to iceberg fluxes of 4.3 and 10 Gmol Fe yr$^{-1}$. However, whilst our mean Fe concentration suggests fluxes at the upper end of previous estimates, which produce large global fluxes relative to other Fe supply mechanisms[16], the high variability in Fe content may have consequences for both Fe transfer to the open ocean and fertilization because 4% of ice sampled contains 91% of the Fe (Fig. 1b). This highly heterogeneous Fe distribution is evident throughout the global dataset, in both the Arctic and Antarctic data irrespective of whether the Icelandic (Jökulsárlón) samples are excluded or not (Fig. 1b). Such a distribution raises questions about the fate of Fe-rich iceberg layers in the marine environment, as these account for the vast majority of the total iceberg-to-ocean Fe flux.

**How efficient is iceberg-Fe-delivery**. Regional Fe budgets and models for the marine environment generally assume that all supplied Fe is equal, i.e. that one unit of Fe delivered from atmospheric deposition has the same fertilizing effect as one unit of Fe delivered from icebergs if both sources overlap in time and space. However, several aspects of glacier-derived Fe sources are poorly constrained and these knowledge gaps may explain why modeled iceberg fertilization scenarios diverge between different model simulations; even when similar, and apparently conservative, mean ice Fe concentrations are used (e.g. refs. [17,18]). The general agreement between our estimate of iceberg TdFe concentration (mean of 9.3 µM) and a methodologically independent estimate of labile Fe within icebergs (mean of 6.8 µM, assuming a sediment load of 0.5 g $L^{-1}$)[4], suggests that the mean Fe content of icebergs is surprisingly well constrained on a global scale.

In addition to the approximate order of magnitude uncertainty in iceberg Fe concentration shown in prior work, sparse information is available on how Fe is distributed across the dissolved-particulate size continuum within ice[25], how the speciation of Fe from ice melt affects its biological utilization in the ocean[35], how changes in Fe distribution and concentration in ice after calving affect Fe delivery[36] and what depth distribution this Fe is delivered over from melting icebergs[18]. Even with improved constraints on the total dissolvable Fe (TdFe) content of ice (Fig. 1a), several unknowns remain in how efficiently this Fe fuels marine primary production and ultimately C export to the deep ocean.

With respect to the lithogenic particle size within ice, several insights can be gained from our dataset. Earlier estimates of iceberg Fe content were deduced using estimated sediment loads within ice with a mean value of ~0.5 g $L^{-1}$, producing iceberg labile Fe contents that are similar to the mean TdFe

concentrations presented herein[4] and to a radium-derived sediment load estimate of 0.6–1.2 g $L^{-1}$ for icebergs in the Weddell Sea[6]. However, the size and distribution of this material within ice is a further cause of substantial uncertainty. Whilst labile Fe content in glacially-derived sediment does not appear to vary substantially with particle size[4,37], large lithogenic particles have shorter residence times in marine waters due to rapid sinking. A sediment size analysis ($n = 51$, Svalbard iceberg samples[38]) suggests a mean particle size of 8.5 µm, which is comparable to the coarse size fraction (~3–10 µm, United States Environmental Protection Agency definition) of atmospheric dust (Fig. 1c). In addition to particle size, the mineral speciation of Fe in particles can affect its solubility and bioaccessibility to marine biota[39]. In this respect, iceberg-derived sediment contains twice as much ferrihydrite, the most labile Fe mineral fraction, as atmospheric dust[4,6,40], potentially making it a more effective fertilizer if delivered in a comparable way to the surface ocean. In summary, solid-state speciation and size analysis suggests that icebergs should be a relatively efficient Fe-fertilizer[4].

A further poorly defined factor is the distribution of Fe within ice. Iceberg Fe concentrations are clearly highly variable (Fig. 1b), with the range of measured concentrations being large compared to other freshwater-associated Fe supply mechanisms. For example, total Fe concentrations in rainwater range over two orders of magnitude from ~10 to 1000 nM[41,42], and river water from ~0.1 to 100 µM[43,44]. The 6-order of magnitude range for icebergs (2 nM–2 mM) reflects the compositional difference between Fe-rich basal ice and non-basal ice, which contains only low concentrations of Fe derived from atmospheric deposition[23]. Such a large range in Fe concentrations within individual icebergs due to this inherent contrast likely explains why regional, or catchment specific, differences in Fe concentrations are not evident in a global dataset of this size (Fig. 1). This heterogeneous distribution underpins the difference between the global 9.3 µM mean and 170 nM median iceberg Fe content, and raises questions about how this affects the lateral and vertical distribution of Fe input into the ocean from iceberg melt. The fertilization potential of particle and Fe-rich iceberg layers may depend on where these layers are located, and at what depth the associated meltwater is released into the ocean.

**Fe release from melting icebergs**. In order to explore how the rate at which the Fe content of iceberg meltwater varies with time after calving, we incorporate Fe content into a model in which iceberg melt rates have been constrained using observational data[45]. Fe fluxes from iceberg melt may vary as a function of iceberg geometry, iceberg spatial distribution, time and environmental drivers (e.g., ocean temperature and salinity, wind speed, air temperature, shortwave radiation flux and sea-ice concentration). As catchment-specific observations are required to compute iceberg melt rates, we use summertime data from Sermilik fjord (Southeast Greenland), the only catchment globally for which iceberg distribution, morphology, size and melt rates have been previously constrained[45,46].

Field observations suggest that icebergs contain sediment-rich layers associated with high Fe content[6,23], which are visible as narrow bands bisecting icebergs or as sediment-coated iceberg faces[23,40,47]. In our iceberg melt model, we consider three idealized Fe distribution scenarios: a random Monte Carlo distribution, a basal-dominated distribution, and a peripheral shell-dominated distribution (Fig. 2). These scenarios are endmembers in the sense that all real icebergs contain Fe distributions that fall somewhere in-between the basal/shell and Monte Carlo approaches. One of the largest uncertainties concerning iceberg melt rates concerns wave-induced melt[48,49]. Wave-induced melt can be parametrized

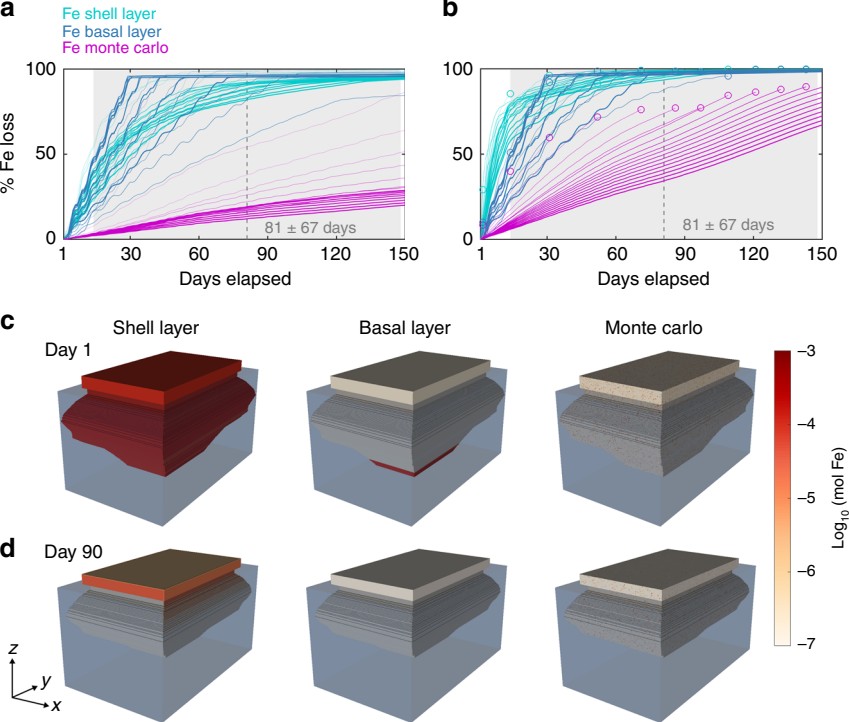

**Fig. 2** Cumulative Fe release from different icebergs. Fe is initially distributed either randomly in a Monte Carlo simulation, concentrated in a layer of basal ice, or concentrated in a shell around the iceberg periphery. Icebergs of varying morphology, based on observed distributions, are released into Sermilik fjord at t = 0 days. **a** Mean net Fe loss from all icebergs within each Fe scenario. Days elapsed represents the model run length with mean iceberg residence time (dashed vertical line) ± standard deviation (shaded gray area) within Sermilik fjord shown for reference[46]. Thicker lines correspond to increasing iceberg length (from 50 to 1000 m). Wave melt is parameterized as previously[45]. **b** The same scenarios, but with an alternative parameterization for wave-melt driven iceberg erosion[50]. Open dots represent icebergs reaching a point of instability where rolling or disintegration becomes inevitable, which would likely accelerate melt and substantially redistribute any remaining shall/basal Fe. **c** Model run at t = 0 days for select shell, basal and Monte Carlo iceberg scenarios. **d** Model run at t = 90 days for the same shell, basal and Monte Carlo iceberg scenarios

as either only affecting the portion of an iceberg near the waterline (as previously[45], Fig. 2a), or as affecting the entire iceberg face[50] (Fig. 2b), with the most realistic formulation presently unclear. If the later approach is used in Sermilik, melt increases substantially over timescales within the typical fjord residence time, with many icebergs reaching a point of instability where inversion or disintegration becomes inevitable (open dots in Fig. 2b).

In both cases, disproportionately high loss of Fe from icebergs occurs within a few days to weeks of calving (Fig. 2a, b). The Monte Carlo distribution, which represents how iceberg Fe fluxes have been estimated and typically modeled to date, results in a 10–70% decline in iceberg Fe content over the mean iceberg transit time of Sermilik Fjord (Fig. 2a, b, magenta lines) depending on iceberg size and how wave-melt driven iceberg erosion is parameterized. This is coincident with a 10–70% decline in iceberg volume. However, the shell and basal scenarios result in 60–99% loss of Fe over the same time period (Fig. 2a, turquoise and blue lines) for the same ice volume loss. This is consistent with the spatial distribution of ice-rafted debris in Sermilik fjord sediment cores[51], which decreases away from Helheim glacier. Fe lost in this near-shore environment is unlikely to enhance primary production, as coastal waters are already fertilized by Fe from runoff and sedimentary sources[52].

Our model simulations demonstrate that the post-calving age of an iceberg has a strong influence on its remaining Fe content, with the rate of Fe loss highly dependent upon the initial Fe distribution (Fig. 2). Further complexities certainly arise between catchments due to differences in ocean temperature, iceberg size distribution, coastal ocean dynamics and uncertainty in the

thickness of basal ice. Basal ice thickness is poorly constrained and therefore here we defined the high Fe layers as occupying 9% of ice volume at t = 0 based on the observed distribution of TdFe (Fig. 1b)[6,36]. Reducing this would amplify the difference between basal/shell and Monte Carlo scenarios (Fig. 2). Given the relatively fast loss of Fe that occurs from basal and shell layers in Sermilik within a few days of calving (Fig. 2a, b), we may still have underestimated the extent of Fe loss in these scenarios because our iceberg samples (Fig. 1) represent ice which calved a few days prior, rather than the marine-terminating glacier ice endmember.

An alternative iceberg melt model aiming to constrain sediment deposition, which should therefore also apply to particulate Fe within ice[6,23,38], similarly estimated that 70–85% of iceberg-borne sediment was deposited within Kangerdlugssuaq fjord (E Greenland)[36]. Nevertheless, both of these model simulations represent Arctic marine-terminating glaciers in fjord systems where icebergs have a moderately long residence time (81 ± 67 days for Sermilik[46]) and are exposed to relatively warm ocean waters (1–4 °C in Sermilik summertime surface waters[45]) before entering the Atlantic Ocean. We therefore consider that, in a global context, our simulated basal and shell scenarios for these conditions (Fig. 2) represent approximate lower bounds for the fraction of Fe entering the Atlantic Ocean from icebergs. Reduced uncertainty concerning the spatial distribution of ice and Fe loss in other catchments will only be possible once further observational data becomes available and Fe/sediment distribution scenarios are better defined. This is especially the case with respect to regional differences in iceberg dimensions and sediment distribution.

Whilst we observe no significant differences in TdFe between catchments herein (Fig. 1), the dataset is biased towards sampling of smaller ice fragments (<20 m length above the waterline) which were originally calved from marine-terminating glaciers rather than large ice shelves. While Fe concentrations within terrestrial ice cores and marine ice (formed at the base of some ice shelves) overlap with the TdFe range we report for icebergs herein[53], critical differences in the distribution of TdFe within recently-calved icebergs may still occur between catchments. This may especially be the case for ice calved from large ice shelves compared to ice calved from inshore marine-terminating glaciers, as the pathways for sediment incorporation into ice (and thus sediment loss from ice) differ between these scenarios and are not well characterized[53,54]. Critical unresolved questions are: how thick are Fe-rich layers, where are they located within different icebergs, how fast are these layers eroded in the marine environment, and how does this vary regionally?

**The depth dependence of melt.** Alongside the distribution of Fe within aging icebergs, the depth distribution of iceberg melt injection is a further interconnected issue widely acknowledged to affect the efficiency of iceberg-Fe delivery to the ocean mixed layer[22,26,45], yet poorly constrained in the global ocean. In Sermilik, 68–78% of summertime iceberg melt is injected into water beneath the shallow (10–20 m) surface layer, with this fraction predicted to remain entirely below the surface throughout summer[45]. However, this finding provides limited insight into the behavior of meltwater around large icebergs in the open ocean, where the mixed layer is deeper and melt rates are generally lower (e.g. Supplementary Note 1 and Supplementary Fig. 1).

Whilst plume theory is reasonably well developed to approximate the behavior of subglacial discharge in 2 or 3 dimensional water columns[55–57], subsurface iceberg melt behavior is less well constrained. Submarine melt rates are imperfectly matched by theoretical calculations[58], and both the non-static nature of icebergs and the dilute nature of iceberg-associated plumes makes them challenging to define. It is a widespread hypothesis that iceberg melt upwells water to the ocean surface, with plumes of melt-modified water then spreading laterally away from icebergs enriched in the micronutrient Fe[7]. Evidence for such upwelling has indeed been observed around icebergs in some cases, yet is also notably intermittent and highly-spatially variable[22,24,59]. The behavior of meltwater in the water column depends strongly on ambient ocean conditions[59], particularly the relative iceberg-ocean velocity. High relative velocities may lead to a detachment of meltwater plumes from the iceberg face and a broader distribution of meltwater through the water column[21].

Idealized buoyancy plume calculations can be conducted for an iceberg in any region where water column properties are defined, and indicate that the fraction of ice melt upwelled to the surface and the fraction of ice melt injected below the mixed layer vary[45] (e.g. see Supplementary Fig. 1). Yet the practical application of this to icebergs in a dynamic ocean is presently limited due to uncertainties with respect to the parametrization of melt rates[58], as noted for wave induced melt (Fig. 2a, b), a paucity of close-to-iceberg data to verify such calculations and the fundamentally weak and intermittent nature of iceberg melt plumes in the ocean. The mixing dynamics between melt and ocean waters remains a key uncertainty in how Fe from ice enters the ocean, and the extent to which it is subsequently made available to microorganisms within the mixed layer[21,22]. We note that, given the heterogeneous distribution of Fe within icebergs, and the variable fraction of subsurface ice melt that may be brought to the surface or into the mixed layer (Supplementary Fig. 1), it is important in

future work to constrain how these two features interact — especially with respect to submerged basal layers.

**Implications for Fe supply to phytoplankton.** Defining iceberg Fe concentration shortly after calving is a bottom-up approach to defining the Fe flux from icebergs by investigating total iceberg Fe concentrations. Equally important insight into the utilization efficiency of iceberg-derived Fe can be gained from a top-down approach, such as by using satellite-derived chlorophyll data in iceberg affected regions[5,8,9]. Given that iceberg Fe concentrations (Fig. 1) suggest iceberg Fe fluxes into the ocean are at the upper end, or greater than earlier estimates, it is perhaps useful to revisit prior attempts to define the efficiency with which iceberg-derived Fe is utilized on a regional scale. Regional Fe utilization was previously estimated as $7–14\,\mu mol\,m^{-2}\,yr^{-1}$ in areas of the Southern Ocean around the Antarctic Peninsula predicted to have a total iceberg derived supply of $72–726\,\mu mol\,m^{-2}\,yr^{-1}$ [5]. In contrast, regions with significant modeled atmospheric dust deposition ($11–38\,\mu mol\,m^{-2}\,yr^{-1}$), such as down-wind of Patagonia, South Africa, and Australia[60,61], had an Fe utilization that matched the modeled atmospheric flux[5]. The apparent difference between Fe supply and utilization for icebergs was attributed to potential temporal mismatches between supply and demand. However the spatial overlap of multiple Fe sources downstream of the Antarctic Peninsula makes it challenging to determine from satellite-derived data alone how efficient icebergs are as a source of Fe to surrounding waters[5,17,18]. Furthermore, satellite-derived chlorophyll data is less reliable at high latitudes[62]. Nevertheless, updating the total Fe flux to values at the upper end of the previously used range would amplify the apparent mismatch. However, this also raises a critical question concerning the heterogeneous nature of Fe in icebergs across any region (Fig. 1b): is it approximately correct to assume that this Fe will be spread evenly across the ocean, or approximately valid to estimate the potential fertilizing effect of icebergs from the total Fe flux or mean iceberg TdFe concentration? Furthermore, it is unclear on how broad a scale Fe-fertilization from icebergs operates. An influence of iceberg passage is generally observed on phytoplankton and nutrient distributions in iceberg tracks during the growth season[9,12,23], but the regional effects of icebergs are more challenging to deduce from satellite derived data alone and are still subject to considerable uncertainties between models[17,18]. Identifying and tracking areas of iceberg Fe-enrichment is inherently difficult and thus any calculations to establish utilization rates are more challenging than for some other Fe sources[5].

Here we have considered only TdFe, which largely consists of labile particulate Fe (>0.2 μm). Dissolved Fe (<0.2 μm) concentrations are far lower in icebergs, and generally not proportional to TdFe, with reported dissolved Fe concentrations ranging from 4 to 610 nM (Weddell Sea)[23], 1 to 540 nM (Svalbard)[38] and <3 to 300 nM (Greenland)[63]. A key factor controlling the extent to which any source of particulate Fe is maintained in more bioavailable dissolved phases, with longer residence times in the water column, is the availability of dissolved organic ligands in seawater[64]. The saturation of ligands in ice-melt affected waters would act to cap the transfer of Fe between the particulate and dissolved phases; this process would limit the fraction of iceberg-derived TdFe available to support primary production on timescales of days to weeks after meltwater enters the water column[65]. In one of the only studies to investigate ligand availability around icebergs, Fe binding ligands were found to be close to saturation in the Weddell Sea[65] and ligands are similarly hypothesized to limit the transfer of Fe into the dissolved phase

downstream of the intense inorganic Fe outflows of large glaciers[66,67]. Such a constraint challenges the assumption that mean, or total, iceberg Fe content can be used to predict broad-scale biological effects because the capping effect of ligand availability could strongly moderate the transfer of iceberg-derived Fe from Fe-rich ice.

We illustrate this by considering a mixing scenario between ice melt and Fe-deficient waters with an excess ligand concentration. Taking the freshwater excess calculated in melt enriched layers in iceberg-affected regions of the Weddell Sea suggests a melt water enrichment of ~0.1%[22]. Assuming, unrealistically, this was uniformly distributed through the surface mixed layer with total measured Fe ligand concentrations in this region ranging 1.2–2.4 nM[65], the maximum quantity of Fe that could be transferred into the dissolved phase in the surface mixed layer would be a freshwater endmember of ~1.2–2.4 μM. This assumes the residence time of ice-derived particles in the mixed layer is short, exchange between particulate and dissolved Fe phases is rapid and that iceberg melt is occurring in Fe-deficient waters. The maximum enrichment possible would be lower in regions where primary production was not Fe-limited, or where multiple Fe-sources overlapped[65], and slightly higher in surface waters due to photochemical processes creating soluble dissolved Fe(II)[65]. For icebergs with a TdFe concentration close to the median (170 nM), a large fraction of the labile Fe present could therefore potentially be transferred to the dissolved phase. However, this declines to only 13–26% for icebergs with TdFe close to the mean (9.3 μM). Such a calculation is over-simplistic in many respects as it neglects to consider the different depths of intrusions and mechanisms of melt occurring in the water column[22]. Furthermore, it is unclear on what spatiotemporal scales iceberg-derived particulate Fe is removed from the water column and how quickly the labile particulate and dissolved Fe phases achieve equilibration. Nevertheless, for ice with high TdFe content, a high fraction of iceberg TdFe could plausibly be maintained in the mixed layer only under

very dilute meltwater addition scenarios. Therefore, the use of a mean iceberg Fe content to represent highly variable iceberg Fe input into the ocean from a point source delivered at the surface is questionable[18].

**Uncertainties in C export from iceberg fertilization.** Given the range of uncertainties, estimates of the extent to which iceberg-derived Fe enhances primary production[7–9] and C export vary significantly[7,17,68]. This is especially the case in the Fe-limited Southern Ocean, where iceberg Fe fertilization is expected to have the greatest stimulatory effect[69], but the relationship between Fe input and C export is complex[70]. The simplest method for estimating how any Fe input into the ocean affects C export, the quantity of POC sinking beneath 100 m depth, is by using Fe-to-C sequestration efficiencies[71,72]. Fe-to-C sequestration efficiencies have been estimated for several naturally Fe-fertilized, high-latitude regions (including the Crozet Islands, Kerguelen Plateau and Irminger Basin[71–73]). Yet Fe-to-C sequestration efficiencies vary widely from 17 to 2900 Kmol C mol⁻¹ Fe and are thus challenging to apply at broader scales[71,73]. Even if they are applied to a specific regional ice melt Fe-enrichment scenario, for example the simplistic ~0.1% melt water enrichment scenario outlined above, a very broad range of POC export is plausible considering the range of iceberg Fe content (Fig. 3a). The mean iceberg TdFe concentration, coupled with an intermediate or high C sequestration efficiency would produce a high C export. Yet a median iceberg TdFe concentration would correspond to C export an order of magnitude lower irrespective of the C sequestration efficiency (Fig. 3a). Additionally, none of these calculated sequestration efficiencies concern regions where icebergs are thought to dominate Fe supply.

An alternative method to convert iceberg Fe input into plausible POC fluxes is to use an estimate of Fe:C cellular ratios to calculate the primary production potentially supported by an

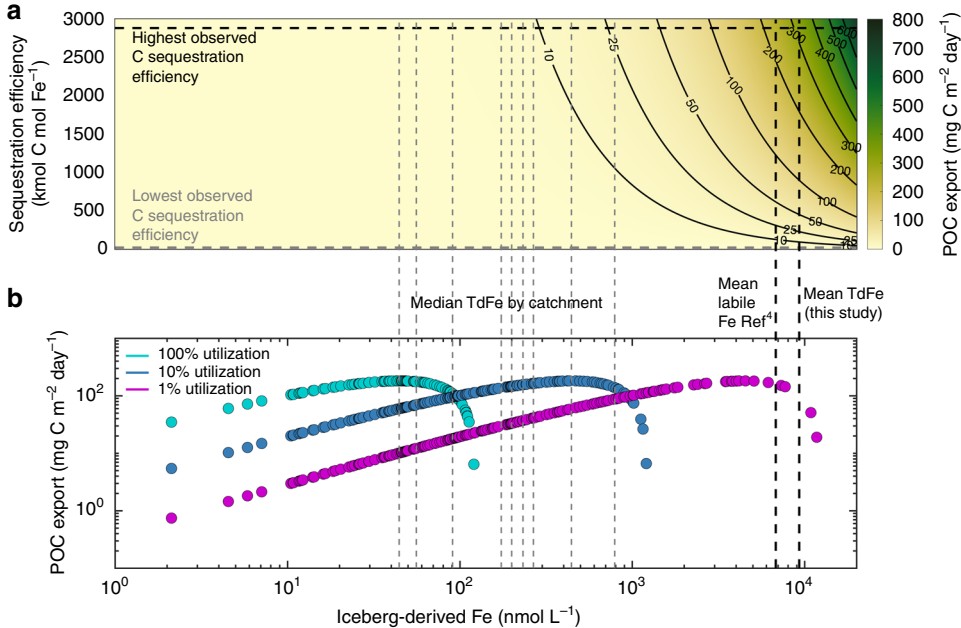

**Fig. 3** Estimating C export fluxes. Following fertilization of Fe-limited waters for a regional 0.1% meltwater enrichment, C export is estimated using two different methods. **a** Fe-to-C sequestration efficiencies estimate the organic C flux to > 100 m depth following Fe fertilization. Vertical lines correspond to the median TdFe of ice each catchment (gray dashed lines), mean labile Fe derived from estimates of iceberg sediment content and mean TdFe for this dataset (black dashed lines). **b** POC export is estimated for 1, 10 and 100% Fe utilization (the percentage of Fe supply up-taken by biota) scenarios. This Fe supply is combined with Fe:C cellular ratios and the observed trend between surface primary production and C export efficiency in the Southern Ocean[74]. Each dot corresponds to a measured iceberg TdFe concentration

**Table 1 Conversion between Fe, primary production (C) and C export, for a specific scenario where a uniform mixed layer of 100 m is fertilized with a 0.1% meltwater enrichment for each iceberg in the global dataset**

| Iceberg fertilization scenarios in the Southern Ocean based on a 0.1% meltwater addition to Fe-limited surface waters | Caps applied | Fe:C cellular ratio $\mu$mol Fe mol$^{-1}$ C | Mean primary production mg C m$^{-2}$ day$^{-1}$ | Mean C export efficiency | Mean C export mg C m$^{-2}$ day$^{-1}$ | Change from baseline C export (a) |
|---|---|---|---|---|---|---|
| a  Empirical relationship between export efficiency and PP[74] | PP | *3* | 2300 | 0.1 | 80 | **-** |
| b  Empirical relationship between export efficiency and PP[74] | Fe | *3* | 2300 | 0.1 | 80 | **0%** |
| c  Empirical relationship between export efficiency and PP[74] | PP | *20* | 1300 | 0.2 | 90 | **+11%** |
| d  Constant export efficiency | PP | *3* | 2300 | *0.1* | 230 | **+190%** |
| e  Constant export efficiency | PP | *3* | 2300 | *0.2* | 460 | **+470%** |

Caps are applied either to primary production (PP, max 3000 mg C m$^{-2}$ day$^{-1}$) or Fe concentration (post-mixing max 2 nM). C export efficiency (the fraction of organic C exported beneath 100 m depth) is limited to the range 0 to 1. Means refer to the average of the calculated response for every iceberg in the dataset- including values in earlier literature ($n = 206$). Values in italics correspond to constant values

Fe flux and then to use the observational trend between primary production and C export to deduce the resulting POC flux (Fig. 3b). The fraction of C formed by primary production which is ultimately exported to depth is however also subject to pronounced spatiotemporal variation. C export efficiency, ranging from 0–1, is defined as the fraction of organic C formed by primary production that is exported below 100 m depth, and observational evidence suggest that this efficiency generally declines with increasing primary production[74]. Calculating the potential primary production supported by an Fe input from Fe:C cellular ratios of 3–20 $\mu$mol Fe mol$^{-1}$ C, again for a specific ~0.1% melt water enrichment scenario for simplicity, with a mixed layer depth of 100 m, and a constant iceberg fertilization effect lasting a month[7,22], the uncertainty in C export efficiency amplifies the uncertainty in iceberg-Fe fluxes into the surface mixed layer (Table 1).

In scaling Fe input to primary production using Fe:C cellular ratios, the effect of applying a cap to primary production (e.g. 3000 mg C m$^{-2}$ day$^{-1}$) or to the maximum stable concentration of Fe in the water column (e.g. 2 nM), approximating upper limits to observed values[23,74], is the same (Table 1a, b). A more significant query is how C export is scaled to primary production. Southern Ocean observational studies, using both sediment traps and thorium-234 based estimates of POC export, suggest a pronounced decline in C export efficiency with increasing primary production in both spring and summer[74]. Yet such a relationship between primary production and C export efficiency is not reproduced in most biogeochemical models (e.g. ref. [18]) or accounted for in some iceberg POC export calculations (e.g. ref. [7]), meaning that differences in estimates of iceberg-Fe-fertilization can arise from how Fe is scaled to C export independently of how the iceberg Fe flux is quantified.

Aside from the general question concerning how C export scales with primary production in the high-latitude oceans[75], a more specific question is whether iceberg fertilized regions will deviate from the background trend in any region. Do icebergs have a unique primary production-C export relationship due to mixing effects, biological community shifts, ballasting effects from lithogenic particles or other features unique to icebergs[12,15,68]? While there is limited field evidence that icebergs locally enhance C export[15,68], there is insufficient evidence to evaluate if the presence of icebergs results in significant changes to regional C export efficiency. It is presently unclear on what spatiotemporal scales iceberg-induced changes to C export operate because

enhanced primary production can lag behind iceberg passage by days to weeks and thus fertilized regions can be detached from the immediate vicinity of icebergs[7,9,12]. Furthermore, in addition to the short-term fertilization following Fe release into the surface mixed layer considered in both ship-based and satellite-based approaches[9,12], increasing concentrations of Fe at depth could have a weaker but more widespread fertilizing effect on longer inter-annual timescales[18].

In summary, the utilization of iceberg-derived Fe in the marine environment is challenging to determine and a broad range of C export scenarios are plausible for any specific Fe addition and ice melt mixing scenario (e.g. Fig. 3). We note that if C export in the high-latitude oceans does not always scale proportionately with summertime primary production (which is what observational studies suggest[70,74]), Fe may more efficiently fuel C export when a dilute Fe addition induces low primary production with high C export efficiency, than when a concentrated Fe addition induces high primary production but with low C export efficiency (Fig. 3b).

With increasing fluxes of solid ice discharge into the ocean, the importance of icebergs as a source of Fe to the ocean is widely expected to increase[11,76,77]. Using global observations, we demonstrate that total Fe fluxes from icebergs into the ocean are already in excess of 14 Gmol Fe yr$^{-1}$. However, iceberg Fe concentrations are extremely variable (Fig. 1b). This directly affects the potential fraction of Fe transferred into the open ocean (Fig. 2), the subsequent distribution of Fe at the point of injection into the ocean and therefore the efficiency with which this Fe can stimulate primary production and C export. In order to reduce uncertainty concerning iceberg ocean fertilization, future parameterizations of this phenomenon in global ocean biogeochemical models must account for heterogeneity in iceberg morphology and Fe content. This will require integration of iceberg-Fe distributions into iceberg melt models. Due to the sensitivity of iceberg Fe delivery to iceberg morphology and internal Fe distribution (Fig. 2), climate-driven changes in the spatiotemporal footprint of iceberg melt may influence marine productivity more than changes in total iceberg Fe flux into the ocean — a feature which is yet to be considered in forecasts of ocean productivity under future climate scenarios.

## Methods

**Ice samples.** Low-density polyethylene (LDPE) sample bottles were pre-cleaned in a three stage procedure (detergent, 1 M HNO$_3$, 1 M HCl) and then stored double

bagged until required in the field. Iceberg samples were collected by hand using metal-free tools. Sample collection was randomized at each fieldsite location by collecting ice samples at regular intervals along pre-defined transects using small boats. Targeted icebergs had visible widths of 0.5–20 m above the waterline. In total 1–5 kg ice pieces were retained in LDPE bags and melted at room temperature. The first 4 aliquots of meltwater were discarded. 125 mL of meltwater was retained unfiltered in trace metal clean LDPE bottles. Trace metal samples were then acidified to pH 1.9 by addition of 180 μL HCl (UPA, ROMIL) and allowed to stand upright for >6 months prior to analysis via inductively-coupled, plasma mass spectrometry (Element XR, ThermoFisher Scientific) after dilution with 1 M $HNO_3$ (distilled in-house from SPA grade $HNO_3$, Roth). Calibration was via standard addition with a linear peak response from 1 to 1000 nM Fe ($R^2 > 0.99$). Analysis of the reference material CASS-6 ($n = 12$) yielded a Fe concentration of 26.6 ± 1.2 nM (certified 27.9 ± 2.1 nM). 51 sediment samples collected from iceberg surfaces or from embedded within icebergs (as per Ref. [38]) and stored frozen were defrosted at room temperature and analyzed for particle size distributions within the size range 0.1–1000 μm using a Laser Analysette.

In addition to previously unpublished data from 152 new samples collected and analyzed herein, existing comparable data was compiled from prior work in Greenland[29,63], Svalbard[38] and Antarctica[3,23,30,31,78] and is included in the global average iceberg composition. Previously published data from Godthåbsfjord ($n = 9$)[63], Kongsfjorden ($n = 28$)[38], South Bay ($n = 7$)[78] and Maxwell Bay ($n = 7$)[78] is included alongside new data in these catchments (e.g. Fig. 1a).

**Iceberg melt model experiments.** To quantify variability in iceberg Fe content and distribution under realistic melting conditions, we used an iceberg melt model[45] constrained by ice, ocean, and meteorological conditions in Sermilik Fjord. Time- and depth-dependent iceberg geometry (i.e., width, length, keel depth, and freeboard height) was used to produce 3–D representations of iceberg volume and Fe concentration at daily time steps. The horizontal and vertical grid resolution was 1 m, with fractional grid cells used for ice volumes < 1 m³. Iceberg lengths from 50 to 1000 m (at 50 m intervals) were simulated; we then scaled the ice volume and Fe concentration by the observed iceberg distribution in Sermilik fjord to compute net ice volume and Fe loss, respectively (Fig. 2).

The dominant decay term in iceberg melt is often wave erosion of the iceberg's vertical faces. Previously using this iceberg melt model[45] the wave erosion term was defined as affecting only an area equal to the wave amplitude above and below the waterline. Over short time periods (<8 days), there is minimal difference between this formulation and one where wave erosion is applied to the entire iceberg vertical face (the difference being included within the uncertainty in ref. [45]). However, for longer time periods which are comparable to the residence time within Sermilik fjord (mean of 81 days), the difference between these two wave-melt formulations becomes larger. Both approaches are therefore considered (Fig. 2a, b). Furthermore, over this longer time period some icebergs, especially towards the smaller end of the modeled range (50–1000 m length) become unstable (defined as length/width/depths which would be likely to capsize and/or initiate disintegration into multiple smaller icebergs). As modeling such instability or fragmentation is challenging and unsupported by field observations, icebergs reaching a point of instability are therefore removed from the model run (as indicated by open dots on Fig. 2a, b).

We tested 3 idealized iceberg Fe distribution scenarios: a shell distribution, a basal-layer distribution, and a Monte Carlo distribution. For the shell and basal-layer cases, high and low Fe ice was set to 100 μM and 170 nM, respectively; grid cells with high Fe constituted 9% of the total iceberg volume representing the 9% of samples with TdFe > TdFe$_{mean}$. For the shell case, the albedo for the iceberg top and above-freeboard sides was reduced from 0.7 to 0.3 when high Fe ice was present (representing the influence of sediment-rich ice)[79]. The Monte Carlo case assigned Fe concentrations, randomly sampled from the observational dataset, to random grid cells until the total Fe content was equal to the shell case; all cells not assigned by the Monte Carlo method were set to low Fe ice. Iceberg simulations started on May 1st and were time stepped for 180 days. Basal layer thickness is poorly defined in the literature with general estimates of <3–10 m[6,36]. Here we used a fractional composition of 9% high Fe for basal/shell scenarios based on the TdFe data distribution. For substantially thinner basal layers, the difference between Basal/Shell and Monte Carlo scenarios in Fig. 1 would be amplified.

**Primary production and C export calculations.** Fe supply to primary producers was calculated as 2× the extrapolated flux from iceberg TdFe concentrations i.e. using a 1/fe ratio of 2 for high Fe waters[5,80]. Except where stated otherwise, a constant cellular Fe:C ratio was fixed as 3 μmol Fe mol$^{-1}$ C[81]. The relationship between primary production and C export was determined using data from Ref.[74] and the fit C export efficiency $= -0.3484 \times \text{Log}(PP) + 1.2239$ with a cap applied at zero export to prevent negative values in C export. Except where stated otherwise, C export is defined at 100 m, and the mixed layer is defined as 0–100 m.

## Data availability

All new data is available in the main text or the Supplementary Materials. Source data for iceberg TdFe concentrations underlying Figs. 1–3 are provided as a Source Data file (Supplementary Material).

## Code availability

The modified code for Fe release from melting icebergs (originally from ref.[45]) is available from https://github.com/dcarrollsci/iceberg_Fe_model.

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

## Acknowledgements

Code for the construction of Fig. 2 was adapted from that originally described in ref. [45], which was kindly made available by the authors. Twila Moon is thanked for comments on the adaptation of the model and Siao Jean Khoo for particle size distributions. Antarctic sampling was possible through FONDAP-IDEAL 15150003 and FONDECYT 3180152. INACH, GINR and Boris Koch/Uwe John (MSM56 cruise) are thanked for logistical support during Antarctic and Greenlandic sampling campaigns.

## Author contributions

M.H., J.H., L.M., L.B., C.E., and H.G. conducted fieldwork and laboratory analysis; D.C. and D.S. undertook modeling work; D.C., F.M., and M.H. produced the figures; M.H., D.C., F.M., and E.A. wrote the manuscript with all authors contributing to its revision.

## Competing interests

The authors declare no competing interests.
