## [Peer Review File · Nature Communications]

Reviewers' comments:

Reviewer #1 (Remarks to the Author):

The manuscript by Hopwood et al. investigates the potential of icebergs to supply ocean waters with the micronutrient iron (Fe). The authors use new measurements of "acid soluble" iron (unfiltered acidified, filtered ~6 months later) from a range of icebergs with a large spatial coverage to determine representative concentrations. This is coupled with modelling of temporal iceberg melt and comparison to Fe utilisation in Southern Ocean waters. The main argument made is that the fertilization potential of icebergs is reduced due to heterogeneous iron distribution. The authors argue that this delivery mechanism is inefficient and may therefore not supply more iron in a warming environment. The manuscript is well written, contains some pertinent information and provides some important arguments about a relatively understudied delivery mechanism. I do however have some issues with it that need addressing before I could recommend publication.

I think the authors are somewhat selective with some of their evidence and with their modelling approach. While I think the use of a simple model to calculate Fe release from icebergs is novel and certainly very useful, I think the scenarios might be somewhat misleading. I understand the use of the basal layer and Monte Carlo scenarios as end members, but struggle to see why the use of a shell scenario is realistic (although I acknowledge it is a thought experiment)? A "middle" more realistic scenario would be one with some internal sediment layers from compression and up-thrust sediment bands (I mention this in my specific comments as well). There is also a recent studies that need to be discussed in this context - Warren et al. (2019). I would like to see these ideas tested and I think it would give value to flux estimates and potentially to some of the major conclusions.

The second major comment I have is the use of Boyd et al. (2012) for Fe utilization rates. I think the value of 7-14 $\mu\text{mol m}^{-2} \text{yr}^{-1}$ is rather pessimistic. Now this might be an average utilization rate across the regions of the Southern Ocean (it's unclear in the manuscript how this was calculated), but it seems too low regions directly influenced by iceberg tracks (the iceberg tracks in Boyd et al. 2012 are somewhat misleading and inaccurate – see <http://www.scp.byu.edu/data/iceberg/> for more accurate tracks). Some parts of the Southern Ocean that are influenced by icebergs have utilization rates far higher (in some locations exceeding 100 $\mu\text{mol m}^{-2} \text{yr}^{-1}$). It may be that I have misunderstood how these are calculated, but if so then this information is absent from the current manuscript. The second point pertaining to these Fe utilization estimates is that these utilization rates are based on Southern Ocean ocean color measurements that are likely to be too low (again see my specific points below – this applies to the Sea-Viewing Wide Field-of-View Sensor's (SeaWiFS) algorithm that is employed in Arrigo et al. 2008, which is used by Boyd et al. 2012). Some papers find chl a are underestimated a factor of 2 or more at latitudes $>60^{\circ}\text{S}$ (Dierssen and Smith, 2000; Guinet et al., 2013; Johnson et al., 2013).

A more general comment is that there is very little discussion of iceberg Fe fertilisation in the context of the two modelling studies mentioned in the text – Death et al. (2014) and Laufkötter et al. (2018). I think it would be beneficial for the authors to contextualize their results in the framework of these modelling studies, mainly because the authors argue for more accurate representation of iceberg Fe fertilization in models.

Specific comments:

- Title: I understand the authors want to be provocative with this, but I my personal preference would be that this is toned down slightly to something like "Highly variable iron content modulates iceberg-ocean fertilisation and carbon export"

- L26-27: I'm not sure the findings allow this conclusion to be reached. The main finding is that heterogeneous distribution limits off shore export, making iceberg fluxes relatively inefficient. However, the authors do not show that an increasing future flux will have little impact on primary production. Again, I understand the desire to be provocative, but this sentence needs toning down

slightly. How about "Future ocean productivity may therefore be less sensitive to increasing iceberg fluxes that previously hypothesized"

- L35-37: But has been included in regional models as detailed in this manuscript (e.g. Death et al., 2014, Laufkötter et al., 2018).
- L62: This is still a lot of iron in the context of iron limited waters (tens of pM), even in relatively pristine ice.
- L74: Good point, but worth noting that the concentrations in the remaining 90% of ice are still relatively high (again in the context of Fe deplete waters).
- L89-93: See my general points. The (uncorrected) algorithms for ocean color are much better in lower latitudes than for the Southern Ocean, which might be introducing some bias into these interpretations, which needs to be acknowledged and discussed.
- L92-93: "Fe utilization that more closely matches the modelled atmospheric flux"
- L93-95: As above - or that utilization rates are incorrect. This uncertainty is not addressed anywhere in the manuscript but is critical for assessing the efficiency of supply mechanisms.
- L119-120: Excellent point but still worth pointing out that 170 nM is still significant
- L133-134: See main comments
- L140-141: Isn't this scenario loaded toward icebergs within fjords (mainly Greenland, Svalbard etc...), where delivery off shelf is limited? This isn't really surprising given the higher basal sediment loadings.
- L165-168: This answers the above point
- L191: Where is this value derived?
- L195-196: I wouldn't call these measurements negligible. Lin et al. (2011) measure concentrations of 0.58-2.92 nM (mean 0.93 nM), which is substantially higher than Fe-limited waters, and would be classified as Fe-replete marine waters.
- L209-215: There seem to be a lot of assumptions used in this calculation with relatively little discussion of uncertainty.
- L214-215: 1% of what? Total flux? Median Fe scenario? Mean Fe scenario?
- L228-231: Isn't this similar to most other delivery mechanisms? Usually the flux at gate is given, without identifying inefficiencies in delivery. Given this, how efficient are icebergs compared to other input sources?
- L237-238: This needs some additional explanation.
- L249-266: Okay – I think this conclusion is nice, but it doesn't really back up the closing line of the abstract. The main findings that I interpret are the a) there are inefficiencies in iceberg transfer of iron to the open ocean, b) that these need to be accounted for in global biogeochemical models, and c) Fe flux does not necessarily imply C export (although I think this is the weakest area of the paper at present). I don't think the argument of little additional iceberg Fe fertilization under future warming scenarios is particularly well substantiated (especially for the Southern Ocean) at present. There is also relatively little comparison to other Fe sources (and the efficiency of those inputs), which I would like to see in order to contextualize the importance of iceberg Fe fluxes.
- Table 1: There's a lot of information in this table which makes it quite difficult to follow. To those less familiar with oceanography literature it would help to clarify some terms (such as mean export efficiency – i.e. fraction of C fixed from NPP exported below 100 m). Also, it is unclear to me how the mean C export is derived from these values. For example, export efficiency and NPP are lower for scenario (e), yet mean C export is higher (compared to (a)-(d)). Some additional explanation would be useful here.
- Figure 3: Could the authors indicate the median Iceberg-derived Fe range here (i.e. 28-960 nM as in the text)?

References:

- Arrigo, K.R., van Dijken, G.L., Bushinsky, S. (2008) Primary production in the Southern Ocean, 1997–2006. *Journal of Geophysical Research: Oceans* 113.
- Boyd, P.W., Arrigo, K.R., Strzepek, R., van Dijken, G.L. (2012) Mapping phytoplankton iron utilization: Insights into Southern Ocean supply mechanisms. *Journal of Geophysical Research-Oceans* 117.
- Dierssen, H.M., Smith, R.C. (2000) Bio-optical properties and remote sensing ocean color algorithms

for Antarctic Peninsula waters. *Journal of Geophysical Research: Oceans* 105, 26301-26312.

Guinet, C., Xing, X., Walker, E., Monestiez, P., Marchand, S., Picard, B., Jaud, T., Authier, M., Cotté, C., Dragon, A.C., Diamond, E., Antoine, D., Lovell, P., Blain, S., D'Ortenzio, F., Claustre, H. (2013) Calibration procedures and first dataset of Southern Ocean chlorophyll *a* profiles collected by elephant seals equipped with a newly developed CTD-fluorescence tags. *Earth Syst. Sci. Data* 5, 15-29.

Johnson, R., Strutton, P.G., Wright, S.W., McMinn, A., Meiners, K.M. (2013) Three improved satellite chlorophyll algorithms for the Southern Ocean. *Journal of Geophysical Research: Oceans* 118, 3694-3703.

Warren, S.G., Roesler, C.S., Brandt, R.E., Curran, M. (2019) Green Icebergs Revisited. *Journal of Geophysical Research: Oceans* 124.

Reviewer #2 (Remarks to the Author):

The manuscript by Hopwood and colleagues explores the role of icebergs as sources of iron to polar oceans. The authors initially approach this topic by presenting an impressive set of field data, documenting iron concentrations of iceberg samples collected from a range of systems. Using this observational dataset in conjunction with modeling of iceberg iron release as a result of melt and carbon export resulting from iron addition, the authors then evaluate large-scale contributions of iceberg-derived iron for carbon export in the Southern Ocean.

The paper's topic is of great general and personal interest, and its submission timely, given the numerous recent results in the physical literature concerning the importance of iceberg and iceberg melt for freshwater fluxes (e.g. Moon et al. 2018, *N. Geoscience*), and the growing body of literature demonstrating ecosystem consequences of ice-ocean interactions. To this reviewer's knowledge this is the most comprehensive survey of iron concentration in icebergs to date, and this aspect of the research itself makes it quite novel. The combination of observational data with modeling of iceberg decay is also an interesting approach, particularly given the extensive supporting data provided by the physical observation collected in Sermilik Fjord.

While I find the observational dataset quite impressive, and the questions being addressed of great worth and general interest, I have qualms with the ultimate scope of the paper, and the approach taken by the authors to link their observational dataset to Southern Ocean carbon fluxes derived from iceberg melt.

1) Whether icebergs can serve as a significant source of iron to the ocean and primary producers is a central question of the paper. Instead of addressing this question, the authors take a particular stance on the issue early on in the paper. A central assumption of the paper, presented in L85-95, is that there is inefficiency in the delivery of iceberg derived iron to primary producers. The authors cite a paper by Boyd and colleagues as supporting evidence for this assumption (specifically, in the case of iceberg supply, values derived from Shaw et al. 2011 for Iceberg Alley), which they state show far greater supply of iceberg iron than biological utilization. Estimates of biological demand / utilizations are in this reference based on a combination of lab and field-derived Fe:C ratios, as well as remote sensing datasets of primary production. Hopwood and colleagues present this mismatch as a fact, which is misleading considering the context in which this data is presented in the original paper. While acknowledging the gap between their estimates of iceberg iron supply and utilization, Boyd et al. make a number of qualifying remarks:

"The rates of iron utilization appear to be considerably less than that potentially supplied from iceberg melt (Figure 6d c.f. Table 2) suggesting some temporal mismatches between supply and utilization. However, determining the extent to which Fe utilization in this region is met by icebergs is not possible due to the confounding effects of overlap with other Fe supply mechanisms such as

Patagonian dust and resuspension of shallow sediments (Figure 6)."

or otherwise, concerning the dataset as a whole:

"The circumpolar estimates of phytoplankton Fe utilization that we produced cannot readily be compared with that for Fe supply, since the latter estimate will be much less spatially resolved (Figure 5). Furthermore, major uncertainties exist regarding the geographical realm of influence of each supply term and in some cases, little is known about putative mechanisms such as Southern Ocean eddies (Table 2). Nevertheless, our initial comparisons between the magnitude of Fe utilization and supply revealed that they are of the same order, and specifically that both regional match-ups and comparisons of maps of regionally distinct Fe utilization with the Fe supply revealed no major mismatches between the magnitude of utilization and supply."

Boyd et al. further acknowledge the large uncertainties in their estimates of biological utilization, stating for example that:

"The sensitivity of Fe utilization estimates to altering either Fe:C ratio or the threshold Fe concentration [...] is greater for the former (Figure 12), with changes in the magnitude of Fe utilization increasing, relative to the standard run, by ~twofold, depending on what set of Fe:C ratios was employed."

While the cited paper therefore claims no statistically significant mismatch considering uncertainties, Hopwood et al. take this difference as a given, basing the rest of their analysis on this premise. It is also worth noting that iceberg affected regions considered by Boyd et al. and in this paper are limited to Iceberg Alley / Weddell Sea, based on remote sensing dataset (sourced from BYU) tracking large (> 5 km) icebergs visible via satellite. Examination of spatial maps of iron utilization in Boyd et al. points to large heterogeneity in biological utilization, with peaks in the West Antarctic Peninsula, Southern Weddell Sea, Amundsen Sea, and Ross Sea, in polynyas or otherwise at the margins of glaciers and ice shelves where icebergs (potentially smaller) would be expected to calve. These sources of icebergs/iron are not considered in Boyd et al. or in this publication (though observations are all collected from smaller systems outside Iceberg Alley), although icebergs are routinely seen in all of these regions.

In general, model based estimates of utilization and availability should be interpreted with caution. For example, the amount of soluble iron derived from aeolian dust the authors quote are, to this reviewer's knowledge, chosen to fit surface ocean iron observations in a modeling / large scale framework, yielding values of 1-10% solubility (an enormous range). Some of this nominal solubility may include abiotic scavenging in the marine environment. The bioavailability of glacier derived iron is a very important topic and one which is deserving of greater attention, via, for example, incubation experiments or seawater additions exploring how much enters the dissolved phase, and how much is taken up in organic biomass. This is beyond the scope of this paper. However, I find the premise of a mismatch a great overreach given context presented in the cited paper, and the use of models without careful discussion of uncertainty, error propagation, or corroborating observations dissatisfying.

2) In examining contributions of icebergs to the environment (i.e. iron concentrations and carbon export), the authors treat icebergs as passive sources of iron, delivering meltwater into the water column without considering the oceanographic implications of this freshwater transport. In this sense their approach is akin to Hawkins et al. (2014), in which a time series of iron concentrations from a proglacial stream is used to derive a flux of iron to the marginal ocean. I emphasize that in both cases the observational datasets underlying the studies are of tremendous significance and importance. However, overarching conclusions concerning impacts on the ocean drawn from such observations should be treated with caution.

In contrast to Hawkins et al., the data in this paper is used to make the opposite point, namely that

the delivery of iron from glacier is inefficient, with numerous reasons for this inefficiency. Several papers, include Hopwood et al. 2018, have considered the impact of freshwater delivery on coastal ocean dynamics, demonstrating that freshwater input serves to impact vertical circulation, with consequences for nutrient fluxes and distributions. A paper cited by the authors in this paper, Lin et al. (2011), also notes that subsurface iron concentrations at the margins of icebergs was elevated, which they attribute to upwelling of iron-rich circumpolar deep water driven by basal melting. Boyd et al. speak of "confounding effects of overlap with other Fe supply mechanisms," which would include upwelling at iceberg margins, or resuspension of sediments both as a result of advection from continental shelves where icebergs are calved but also iceberg grounding along the continental shelf, evidence for which is presented in the geological literature and is particularly relevant for large tabular icebergs in Iceberg Alley.

Icebergs shouldn't simply be considered as passive sources of freshwater to the marine environment, if sweeping claims as to their contributions (or lack thereof) to polar ocean productivity and carbon export, and the Southern Ocean in particular, are made. Interactions between icebergs and oceans should be considered. However, Hopwood et al. make no mention of feedbacks between melt and circulation. At a minimum this source of uncertainty should be acknowledged. In fact, an interesting development of this study would be to use the melt model discussed in L121-169 to explore vertical nutrient fluxes at iceberg margins associated with ocean melting of iceberg, given the extensive physical data for Sermilik fjord, an important freshwater export pathway for the Greenland Ice Sheet. Hypotheses could be generated, and measurements to test these hypotheses could be collected.

3) The paper progresses from discrete observations collected from a range of systems and icebergs across the globe, to a discussion of iceberg melt and iron loss based on a particular Greenland fjord system (Helheim / Sermilik), to an analysis of carbon export in the Southern Ocean. In aggregate these vignettes are designed to support the premise of inefficiencies in the contributions of iceberg to ocean carbon export and iron delivery. The extent to which conclusions from one section relate to the next is however unclear in my mind. For example, hydrography in Helheim / Sermilik shows a warm (~ 4 °C) water mass at depth with a cooler (~ -1 °C) water mass above. This water mass distribution is not representative of what is found in the Southern Ocean, where the warmest water mass, Circumpolar Deep Water, reaches a maximum of ~ 2 C offshore, and significant decreases in temperature are noted across continental shelves to the glacial margins. Circulation patterns in coastal bay / fjord systems are also quite different in Greenland v. Antarctica, with currents in Sermilik reaching tens of cm/s, while such velocities (along with temperature a major determinant of melt rates) are generally not found in coastal fjords of the Antarctic Peninsula, for example. Water masses found along the Western Weddell Sea shelf are generally far colder than along the West Antarctic Peninsula, with icebergs frequently grounded or otherwise circulated along with sea ice. All these factors would impact melt rates, iron loss, and thereby spatial extent of iron inputs, although they are at best glossed over in this article.

Considering an iceberg as a system, its impact will inevitably depend on its age, as the authors discuss, as well as the distribution of iron within icebergs, which remains unknown given that only marginal iceberg samples have ever been collected. However, it should also depend on what particular system the iceberg traverses. The authors largely discount "hotspots" of productivity as contributors to carbon export (L258) based on a parametrization of carbon export, despite observations of the importance of such systems from process studies, and of episodic pulses of production, for carbon export and marine ecosystem processes. Gerringa et al. (2012), and Annett et al. (2017), as well as references therein and citing papers, speak of episodic iron limitation along coastal Antarctica, as well as rapid utilization of glacially derived iron by primary producers. Examining these process from a mean / steady state perspective, using even regionally tuned (remote sensing-based) models, likely hides variability which would have significant impact on the conclusions presented in this study. The topic of carbon export, and its parametrization from large scale (model, remote sensing) observations is the subject of significant ongoing study, including the NASA EXPORTS project. At a minimum, caveats should be presented regarding uncertainties in the model used to derive primary conclusions,

and mismatches between export calculations from large-scale datasets v. in situ traps.

I reiterate that this paper presents very interesting data that contributes significantly to our understanding of iron concentration present in glacial ice. The paper also presents intriguing model results concerning the contributions of iron to the marine environment, and timescales over which this iron is lost, which provides insight into the spatial extent of its potential impact and results in testable hypotheses. Their research, as summarized in L226-228, provides valuable insight into processes that may significantly impact iceberg contributions to the ocean. However, I find the paper, in aggregate, overreaches in its interpretation of models given the observational data, in order to fit the narrative that icebergs are an inefficient source of iron, which is at best a conjecture and not as this paper presents a given. In this respect their concluding paragraph (L258-266) is telling, as they urge better parametrizations for models based on their work, but make no mention of the need for field experiments and observations (process studies, incubation experiments) to test their hypotheses regarding iceberg iron distribution, bioavailability, timescales of iron loss from icebergs, or carbon export resulting specifically from glacial iron input.

There are a number of interesting ways to develop this study, including a broader discussion of under what conditions, for a range of systems (i.e. the ones that have been sampled), icebergs may or may not be an important source of iron, or using a melt model as presented in the second section whether ocean sources of iron, sourced from upwelling, are significant compared to what is exported from the iceberg itself for the particular system where data was collected. One could also explore timescales of iceberg transit and melt in the Southern Ocean from available datasets, with model parametrizations provided by large-scale models (e.g. SOSE). This datasets will make a significant contribution to the field. Simplifying this manuscript to focus on a particular aspect of the story and hypotheses, while giving a broader perspective on the question of iceberg contributions, will make for a stronger discussion.

Minor comments:

L46-48: The sentence discusses fluxes, but a concentration is presented.

L59-61: additional details as to this analysis need to be presented in the methods. for example, a multiple comparison test was presumably run to derive this result, although this is not detailed anywhere. was the ANOVA run on log-transformed data?

L96-98: If icebergs are characterized by extremes in concentrations, with regional variability, and time variation in concentration, a mean concentration is likely an uninformative quantity, particularly given the skewed nature of the distribution of iceberg iron concentrations (e.g., as seen in the distribution of log₁₀ concentrations presented in Figure 1). It's worth remembering that the iceberg samples under consideration, while (again) of tremendous significance, represent only a small subset of all icebergs, that samples were collected only from the outer portion of any given iceberg, and that large tabular icebergs characteristic of Antarctica / Iceberg Alley were not sampled, based on the iceberg dimensions and locations presented in the manuscript.

L193-194: this is in contradiction to the results of the ANOVA discussed in L59-61

L271-273: It's unclear from the text how the described method yields randomized data collection, given that this sentence speaks of "regular intervals" and "predefined transects". Were transect starting points and headings randomized? how long were transects? Were any icebergs omitted from sampling due to safety concerns or other?

Reviewer #3 (Remarks to the Author):

This work by Hopwood et al is novel and extremely timely. Many researchers speculate that the predicted increase in the frequency of iceberg calving events will result in enhance Carbon export, especially in the Southern Ocean. This is because the Southern Ocean is generally anemic and that icebergs have high iron content. Icebergs can therefore become "hotspots" of biological activity. The

paper claims that the general view that iron is distributed homogeneously within icebergs has led to skewed estimates of icebergs fertilisation potential, and therefore carbon sequestration. I find particularly outstanding the wealth of observational data collected in this study. 210 icebergs samples, spread across both poles, is unprecedented. Results are clear, showing extremely large gradients (6 orders of magnitude) in iron content. The field data is then complemented with a model to evaluate the effect of iceberg melt waters, considering 3 different iron distribution scenarios. The age of the icebergs appears to strongly influence the fertilisation potential of icebergs, which makes sense, but has not explicitly been put forward nor studied before. The study does on to then evaluate carbon sequestration under different scenarios. The paper is beautifully written. The introduction is clear and the methods are adequate. Again, collecting pieces of icebergs in the most remote and challenging parts of the world ocean, under trace metal clean conditions, is commendable. I find the section on the inefficiency in iceberg-iron delivery especially convincing. It gives a clear path to why this work is needed. The main findings are very well supported by the field and model results. This paper will undoubtedly be highly cited in the field of chemical and biological oceanography, especially for researchers working in polar waters.

Delphine Lannuzel

Reviewer #4 (Remarks to the Author):

Review of "Highly variable iron content limits iceberg-ocean fertilisation and carbon export" by Dr Hopwood and co-workers

This is my first review of the manuscript, and it will focus particularly on the iceberg modelling approach. The authors present a novel collection of more than 200 global iceberg samples, which shows that iron concentrations within icebergs vary over six orders of magnitude. Using an iceberg melt model applied to an Arctic fjord during summertime, the authors proceed to show that the spatial variability/heterogeneous spatial distribution of iron within icebergs (e.g. concentrated in the basal layer, or as a shell layer surrounding the iceberg) would lead to a rather rapid loss of iron within a few weeks of calving, thus limiting the delivery of iron to the more remote areas of the Southern Ocean, where iron-limitation tends to be stronger than close to the continent. This will therefore also likely limit carbon export. The rapid loss of iron within a few weeks after calving appears to be supported by an analysis (in a previous study) of ice-rafted debris in a sediment core from an Arctic fjord.

The findings of the authors are novel and the fertilization effect of icebergs and the associated potential negative climate feedback are still poorly constrained and under debate. The study by Dr Hopwood and co-workers is therefore timely and of large interest, as the authors also conclude that climate-driven future changes of the iceberg meltwater pattern might impact C export to a larger degree than the potential increases in the total iceberg iron flux itself (which is expected to increase under global warming). This highlights new model development directions for the climate modelling community. The paper is written clearly and the quality of the figures is very good. I therefore recommend minor revisions as detailed below.

Thomas Rackow

=====

General comments:

1) One concern is that "the fraction of iceberg-derived Fe that can be transported to the open ocean" (I.50/51) has been constrained using conditions within a single Arctic fjord during summer, and these results are further used to quantify fertilization and C export "in the Southern Ocean" (I.52). This is

stated once more in lines 231-234. I am aware of the data situation, and the authors state that "reduced uncertainty concerning [...] Fe loss in other catchments will only be possible once further observational data becomes available", but the paper would benefit from a short discussion why these Arctic results are potentially transferable and applicable to the Southern Ocean conditions.

- l. 170ff: It wasn't clear to me how exactly the Arctic model results (in Fig.2) enter the discussion for the Southern Ocean case (in Table 1 and Fig.3)?

- l.74: "4% of the ice contains 91% of the Fe"; This is true for the global dataset. Because of potentially different mechanisms/iron sources in the Northern and Southern Hemisphere, do you get different answers if you do this for Greenland (Grn) and Antarctic (Ant) icebergs (Maxwell Bay, South Bay, maybe Pia fjord) separately? In particular, Fig.1c would be very interesting to see for the "Ant" case only.

- Fig.2 c/d: Regarding the "shell-layer" case, does this case translate to the Antarctic at all? I have seen photos of small "dirty" Arctic icebergs covered by some type of shell layer, but I am not aware of larger Antarctic equivalents of this type. My understanding is that a large percentage of iceberg mass in the Southern Ocean calves as giant icebergs from a handful of large ice shelves around Antarctica, and they might distribute iron/sediment which the ice gathered (at its base) on its route to the coast or by scraping material off the continental shelf ("basal case"). Iron-rich "narrow bands bisecting icebergs" (l.132) have also been observed (not considered here as a separate case), i.a. from atmospheric deposition. A major number of rather pristine smaller icebergs is also generated by disintegrating giant tabular icebergs in the open Southern Ocean.

- l. 158/159: "Further complexities certainly arise between catchments due to differences in ocean temperature, iceberg size distribution, and coastal ocean dynamics" This might be a good place to discuss the many differences when moving to the Southern Ocean in particular. Similar for lines 165-168. As an example, basal melting of giant icebergs might only significantly increase as soon as icebergs leave the coastal current and cross the southern ACC front (Rackow et al. 2017), which could imply a less rapid iron loss (assuming the 'basal case') and higher iron delivery to more remote open-ocean areas.

2) Another minor concern is the description of the iceberg melt model itself, which as I understand has been used in a previous high-profile study, but nevertheless I'd recommend to provide some more details on the model and on the underlying assumptions for better reproducibility of the results. The authors refer to the Methods section and to the paper by Moon et al. 2019 ("Subsurface iceberg melt key to Greenland fjord freshwater budget"), which features an extensive Supplementary Information with more information about the iceberg model.

Although it appears to me that the standard melt parameterizations, known from large-scale modelling studies with much simpler idealized iceberg geometries (Bigg et al. 1997, Gladstone et al. 2001, Silva et al. 2006, Rackow et al. 2017, Stern et al. 2016, Merino et al. 2016, ...), are implemented in your model, I'd appreciate if the basic details/equations of the iceberg model would be given in a Supplement or in the Methods section. Any differences to the model used in Moon et al. (2019) should be explicitly mentioned as well.

- In previous iceberg modelling studies mentioned above, the wave erosion melt rate has often been applied to the whole side area of an iceberg; from the Supplementary Information to Moon et al. (2019), here it is only applied to a narrow strip along the waterline. Why is that? My understanding is that the parameterization accounts for the calving of overhanging slabs as well, and that's why the whole side area is usually used. Moreover, erosion is often the largest decay term in iceberg melt studies, so the model results could be rather sensitive to the choice of the side area.

- Following Nature's recommendations to authors

(<https://www.nature.com/authors/policies/availability.html#code>), it would also be a good idea to share the iceberg melting code on a public repository. Given the structural uncertainty of current iceberg models, which is also mentioned in the Supplement to Moon et al. (2019), a tagged version of the code that was used in your study would greatly facilitate future research that wants to build upon your efforts.

=====

Line-by-line comments:

I.23: "Using a global analysis of iceberg samples" It was not clear to me whether this is a compilation of previously existing data or whether a large part of the collection has been newly done for this study

I.36/37: "and has yet to be explicitly parameterized in global ocean biogeochemical models" One could go as far to say that it has yet to be explicitly simulated in models, e.g. by combining biogeochemical models with Lagrangian models of iceberg drift/decay (as in the references above). (similarly I. 259)

I. 117/118: It appears to me from these two lines that there is a known compositional difference between iron-rich basal ice and non-basal ice of lower iron content. Would you therefore consider the "basal layer" case (I.134 and Fig.2) to be the default, most realistic case of the three?

I. 133: What does "endmember" refer to? Delete?

Fig.2, I.164: "Days elapsed"/"residence time": Iceberg drift is not explicitly modelled, as far as I understand, so that model icebergs will never leave the fjord? So the x-axis shows rather a potential "residence time" in the fjord and just refers to the "length of the model run" in days?

I. 163, I. 205: Delete "endmember"

=====

References mentioned above:

Bigg, G. R., Wadley, M. R., Stevens, D. P., and Johnson, J. A. (1997), Modelling the dynamics and thermodynamics of icebergs, *Cold Reg. Sci. Technol.*, 26(2), 113– 135, doi:10.1016/S0165-232X(97)00012-8

Gladstone, R. M., Bigg, G. R., and Nicholls, K. W. (2001), Iceberg trajectory modeling and meltwater injection in the Southern Ocean, *J. Geophys. Res.*, 106(C9), 19,903– 19,915, doi:10.1029/2000JC000347

Silva, T. A. M., Bigg, G. R., and Nicholls, K. W. (2006), Contribution of giant icebergs to the Southern Ocean freshwater flux, *J. Geophys. Res.*, 111, C03004, doi:10.1029/2004JC002843

Rackow, T., Wesche, C., Timmermann, R., Hellmer, H. H., Juricke, S., and Jung, T. (2017), A simulation of small to giant Antarctic iceberg evolution: Differential impact on climatology estimates, *J. Geophys. Res. Oceans*, 122, 3170– 3190, doi:10.1002/2016JC012513

Stern, A., Adcroft, A., and Sergienko, O. (2016), The effects of Antarctic iceberg calving-size distribution in a global climate model, *J. Geophys. Res. Oceans*, 121, 5773– 5788, doi:10.1002/2016JC011835

Merino, N., Sommer, J. L., Durand, G., Jourdain, N. C., Madec, G., Mathiot, P., and Tournadre, J. (2016), Antarctic icebergs melt over the Southern Ocean: Climatology and impact on sea ice, *Ocean Modell.*, 104, 99– 110, doi:10.1016/j.ocemod.2016.05.001

=====

Dear Editor and Reviewers,

The reviewers are thanked for detailed and constructive comments on the original text. Enclosed please find a revised and restructured manuscript following reviewers' comments. Line numbers in responses to reviewers' comments refer to the revised text (R1). As the complete comments from 4 reviewers are quite lengthy, I have also included a short summary below. References referred to in replies are included at the end.

Quantifying iceberg Fe content. The presentation of our data is subject to minor changes in R1. We show explicitly that the heterogeneity in iceberg Fe content is independent of the region/catchment studied and included more detailed analysis of this in Supplementary material and Figure 1.

Modelled iceberg Fe fluxes and the efficiency of iceberg-Fe delivery. Instead of opening the discussion around the topic with the reliance on a comparison of iceberg supply and utilization (which relies strongly on the use of satellite derived chl a data), we highlight the very different modelled results obtained for iceberg fertilization between independent biogeochemical models. We note that practically identical iceberg Fe endmembers produce very different fertilization responses in state-of-the-art models (eg contrast Refs^{1,2}) raising questions about how insightful these results are. Throughout we have attempted to highlight areas of outstanding large uncertainties where further work is certainly required.

Fe release from melting icebergs. We now extensively discuss existing uncertainties on how ice melt is parametrized in a water column and, in the absence of a definitive 'best' formulation in the literature, opt to use two different formulations for wave-melt. Additionally, we show in simple terms how the temperature difference between different ocean regions (contrasting S Greenland and the Antarctic Peninsula) would impact ice melt rates and ice melt plume buoyancy (Supplementary Material) and discuss the implications of this for Fe supply into the ocean. We also note the extensive uncertainty with how basal layers in particular interact with the dynamic ocean and raise a list of several issues that need to be resolved in further work before icebergs can be realistically characterized in terms of their effect on Fe availability in a dynamic water column.

The potential role of icebergs in ocean fertilization. We simplify the discussion of how ice could fertilize the ocean by considering a reduced number of simplified scenarios and raising several unknowns, the most critical of which are how melt from Fe-rich basal layers enters the water column, and how efficiently iceberg-fertilization sequesters carbon beneath the surface mixed layer. Neither of these issues can be resolved using presently available data and we suggest several specific areas to be addressed which may reduce the present large range of iceberg-fertilization presently indicated by model scenarios.

Reviewer #1 (Remarks to the Author):

The manuscript by Hopwood et al. investigates the potential of icebergs to supply ocean waters with the micronutrient iron (Fe). The authors use new measurements of “acid soluble” iron (unfiltered acidified, filtered ~6 months later) from a range of icebergs with a large spatial coverage to determine representative concentrations. This is coupled with modelling of temporal iceberg melt and comparison to Fe utilisation in Southern Ocean waters. The main argument made is that the fertilization potential of icebergs is reduced due to heterogeneous iron distribution. The authors argue that this delivery mechanism is inefficient and may therefore not supply more iron in a warming environment. The manuscript is well written, contains some pertinent information and provides some important arguments about a relatively understudied delivery mechanism. I do however have some issues with it that need addressing before I could recommend publication.

I think the authors are somewhat selective with some of their evidence and with their modelling approach. While I think the use of a simple model to calculate Fe release from icebergs is novel and certainly very useful, I think the scenarios might be somewhat misleading. I understand the use of the basal layer and Monte Carlo scenarios as end members, but struggle to see why the use of a shell scenario is realistic (although I acknowledge it is a thought experiment)?

We are not sure this is the case. The scenarios are ‘endmembers’ i.e. all realistic Fe distributions would fall somewhere between a Monte Carlo and a shell/basal scenario. None of the 3 scenarios explicitly represents a ‘real’ iceberg. The shell scenario was interesting to test for two reasons. First, the immediate question with a basal scenario is; what if the layer which is initially basal ‘flips’, i.e. a basal layer becomes a side-layer. Second, a more physically orientated question is how the different components of melt –comparing for example shear along the iceberg sides, basal and surface melt–interact with different spatial orientations. From a physics perspective, the cross-over between the shell and basal scenarios is interesting and perhaps not intuitive to scientists in other disciplines.

A “middle” more realistic scenario would be one with some internal sediment layers from compression and up-thrust sediment bands (I mention this in my specific comments as well).

Yes we entirely agree this is a ‘best guess’ for what a real ‘average’ iceberg looks like, and our first attempt at a model was actually for this scenario. But it proved challenging to parametrize internal sediment layers as there is absolutely nothing (that we could find) in the literature on this. The best/only quantitative information we have is the use of this dataset to derived an estimate of the percentage volume of ‘high Fe’ and ‘low Fe’ ice, and upper limits to basal thickness from visual observations^{3,4}. <3 and <10 m were the only attempts to innumerate basal layers we could find in the literature^{3,4}. For a randomly orientated internal Fe-rich layer, of unknown thickness, the Fe release with time would be very sensitive to the orientation and thickness chosen and thus, in our opinion, an exercise of limited use. To run any such scenario effectively we would have to run a large number of layers with randomized ‘sediment rich’ orientations in 3D. However, in any case, plotting the upper and lower limits to such a scenario is basically the same as the Basal and Monte Carlo scenarios which is why we opted for this approach—we now explain this in lines 171-173.

There is also a recent studies that need to be discussed in this context - Warren et al. (2019). I would like to see these ideas tested and I think it would give value to flux estimates and potentially to some of the major conclusions.

- L133-134: See main comments

This text (referred to⁵) concerns marine ice, not glacial ice, which accounts for a small but poorly defined fraction of ice calved around large, ice shelves. The mechanisms concerning Fe incorporation into marine ice are fundamentally different from those of glacial-ice, and could be argued to have more in common with sea-ice. In any case we are not sure that we can add much to this discussion using our dataset which concerns glacial ice. We note that the very few datapoints available for marine ice in the literature have been used to suggest a much higher Fe concentration than non-marine ice⁶, but that within our dataset such a comparison would not be statistically valid – marine ice PFe would be above the median, but close to the mean for iceberg TdFe. The presence of marine ice on the bottom of ice shelves adds to the complexity concerning how basal ice melt affects Fe distributions and is now mentioned briefly accordingly (lines 231-241).

The second major comment I have is the use of Boyd et al. (2012) for Fe utilization rates. I think the value of 7-14 $\mu\text{mol m}^{-2} \text{yr}^{-1}$ is rather pessimistic. Now this might be an average utilization rate across the regions of the Southern Ocean (it's unclear in the manuscript how this was calculated),

It is unclear which manuscript is referred to here, the value is taken directly from Boyd et al. (2012), it is a regional average as per other values in the same paper used for comparative purposes. It should be noted however that at the time this (2012) text was written, the range of iceberg Fe concentrations used to derived mean Fe input from icebergs was very broad because this had to be derived from a best guess of iceberg sediment load (Table 2, Boyd et al., 2012). Using an updated estimate for the iceberg endmember from our dataset would increase any relative difference between apparent utilization and supply because the measured mean from the new dataset is above the range previously used. A major point of our text however is that such an approach is likely not valid because any 'mean' iceberg content largely reflects sediment inclusion which is not likely delivered in a heterogeneous way across the ocean surface (new lines 294-298).

but it seems too low regions directly influenced by iceberg tracks (the iceberg tracks in Boyd et al. 2012 are somewhat misleading and inaccurate – see <http://www.scp.byu.edu/data/iceberg/> for more accurate tracks). Some parts of the Southern Ocean that are influenced by icebergs have utilization rates far higher (in some locations exceeding 100 $\mu\text{mol m}^{-2} \text{yr}^{-1}$). It may be that I have misunderstood how these are calculated, but if so then this information is absent from the current manuscript.

It is clarified that this is a mean for a specific region. The 2012 paper, as far as we can tell, didn't aim to plot iceberg tracks, it aimed to plot an area where iceberg associated Fe should hypothetically be the dominant source. The area used (in 2012) is within the high intensity iceberg track distribution shown in the link provided. There is of course variation in any utilization rate, spatially and temporally, this applies to all Fe sources. The point of the 2012 text, and the calculations we outline herein, is to see how important icebergs could be for broad-scale regional Fe supply. Following specific exact iceberg tracks and then deducing the chl a change is an entirely different approach which is what the reviewer appears to be advocating for, which has been attempted in several different studies with varying results⁷⁻⁹.

The second point pertaining to these Fe utilization estimates is that these utilization rates are based on Southern Ocean ocean color measurements that are likely to be too low (again see my specific points below – this applies to the Sea-Viewing Wide Field-of-View Sensor's (SeaWiFS) algorithm that is employed in Arrigo et al. 2008, which is used by Boyd et al. 2012). Some papers find chl a are underestimated a factor of 2 or more at latitudes >60°S (Dierssen and Smith, 2000; Guinet et al., 2013; Johnson et al., 2013).

This is indeed an ongoing discussion in the use of satellite derived chlorophyll a, especially at high latitudes. But, in the specific case cited the authors were comparing the utilization of different Fe sources at similar latitudes, (and we note that in other cases, supply and utilization match relatively well), thus the absolute value of chlorophyll a is less important than the relative difference between regions with different fertilization mechanisms. Never-the-less the uncertainty in high-latitude chlorophyll a is explicitly flagged (new lines 290-293), although we note that even a factor of two increase wouldn't change the calculated utilization in the 2012 manuscript that much given that the updated total Fe supply would now be at the upper end of the order of magnitude range quoted originally.

A more general comment is that there is very little discussion of iceberg Fe fertilisation in the context of the two modelling studies mentioned in the text – Death et al. (2014) and Laufkötter et al. (2018). I think it would be beneficial for the authors to contextualize their results in the framework of these modelling studies, mainly because the authors argue for more accurate representation of iceberg Fe fertilization in models.

The major comment we have concerning any iceberg model study, is that model results are yet to be reproducible between studies which somewhat questions how useful they are in a biogeochemical context. It is notable that comments to the effect of 'icebergs are efficient ocean fertilizers' appear in the literature, frequently using the highest estimates of fertilization e.g. ^{2,7} and neglecting to mention that other studies using the same methods in similar regions report far lower fertilizing effects e.g. ^{1,8}.

A recent text is particularly useful in addressing this concern¹. Contrasting the Laufkötter et al. (2018) manuscript with a similar alternative model¹ it is apparent that the differences between the two in terms of primary production or C export (8% vs 30% of Southern Ocean POC export) are absolutely nothing to do with the Fe input from icebergs as both models use similar Fe inputs, but produce very different C export results. (Actually, both use a range of inputs which are meant to cover low, medium and high Fe input scenarios, but the outcomes in terms of productivity don't overlap between the two studies). This is why we raise the issue of scaling primary production to C export in the text, because it is apparent that this (rather than the Fe endmember used in model scenarios) is a key reason why some model results and calculated iceberg-C fluxes diverge which is completely independent of the Fe iceberg endmember used (lines 372-380).

We agree it's important to raise this issue and have reworked the text to highlight these differences and to briefly discuss why they are important in a biogeochemical context (new lines 42-45, 110-116, 372-376.)

Specific comments:

- Title: I understand the authors want to be provocative with this, but I my personal preference would be that this is toned down slightly to something like "Highly variable iron content modulates iceberg-ocean fertilisation and carbon export"

Yes this seems an accurate summary of the revised paper. 'Highly variable iron content modulates iceberg-ocean fertilisation and potential carbon export'.

- L26-27: I'm not sure the findings allow this conclusion to be reached. The main finding is that

heterogeneous distribution limits off shore export, making iceberg fluxes relatively inefficient. However, the authors do not show that an increasing future flux will have little impact on primary production. Again, I understand the desire to be provocative, but this sentence needs toning down slightly. How about “Future ocean productivity may therefore be less sensitive to increasing iceberg fluxes that previously hypothesized”

Agreed. Although one piece of literature does support such a hypothesis (all be it, only for one specific catchment). Sediment in ice is a reasonable proxy for TdFe, as the two are directly proportional, and an alternative sediment load model developed for a large Greenlandic fjord suggested that iceberg dynamics were more important than climate forcing for determining the fate of sediment (see new lines 218-220)⁴. In any case we have added new sections discussing general unknowns in our understanding of iceberg-ocean interactions (new lines 120-127, 242-275). We accept that we do not explicitly show that this is the case for future changes in iceberg fluxes, but suggest that it is likely an important consideration given the many factors that affect Fe delivery from icebergs into the surface offshore ocean (lines 428-432).

- L35-37: But has been included in regional models as detailed in this manuscript (e.g. Death et al., 2014, Laufkötter et al., 2018).

Included yes, but not with reproducible results between models^{1,2}. Note these two state-of-the-art models/references with completely different outcomes in terms of POC export.

- L62: This is still a lot of iron in the context of iron limited waters (tens of pM), even in relatively pristine ice.

Yes the number is large, but cannot be compared to saline concentrations without considering the typical dilution in an iceberg ‘plume’. A 100 nM freshwater concentration would lead to a 0.1 nM enrichment in surface waters with a 0.1% meltwater content. Meltwater fractions are typically similar to, or lower than this, and this refers to total dissolvable Fe, not dissolved Fe- which would be lower again. In most cases for our dataset, the ‘enrichment’ from meltwater in the open ocean would be barely detectable- which is consistent with the limited field studies that have attempted to circle icebergs and found the Fe enrichment to be spatially intermittent¹⁰.

- L74: Good point, but worth noting that the concentrations in the remaining 90% of ice are still relatively high (again in the context of Fe deplete waters).

Yes, but also see above.

- L89-93: See my general points. The (uncorrected) algorithms for ocean color are much better in lower latitudes than for the Southern Ocean, which might be introducing some bias into these interpretations, which needs to be acknowledged and discussed.

We add a note to highlight this widespread issue (line 292-293), but as above note that this isn’t as much of an issue for a comparison of chl a in different regions at similar latitudes (which is the context here).

- L92-93: “Fe utilization that more closely matches the modelled atmospheric flux”

Amended.

- L93-95: As above - or that utilization rates are incorrect. This uncertainty is not addressed anywhere in the manuscript but is critical for assessing the efficiency of supply mechanisms.

We have added some discussion around the limitations of satellite derived utilization within the text (lines 277-298).

- L119-120: *Excellent point but still worth pointing out that 170 nM is still significant*

Depending on the dilution factor as outlined above, whether TdFe is going to result in a significant Fe input or not into seawater cannot really be determined without looking at how this material behaves in the marine environment. In some contexts particulate Fe addition actually decreases dFe availability because the scavenging effect ‘outdoes’ the dissolution effect during the residence time of particles in the surface mixed layer. This hasn’t been tested for icebergs but has been explicitly demonstrated for atmospheric dust and ash-both of which contain ample ‘labile’ Fe^{11,12}. We raise this general uncertainty in new lines 294-214.

- L140-141: *Isn’t this scenario loaded toward icebergs within fjords (mainly Greenland, Svalbard etc...), where delivery off shelf is limited? This isn’t really surprising given the higher basal sediment loadings.*

- L165-168: *This answers the above point*

Yes. For extra clarity we have rephrased the sentence in question to read ‘within this catchment’

- L191: *Where is this value derived?*

It is taken directly from the estimates of total meltwater content in upper water column calculated by Ref¹³.

- L195-196: *I wouldn’t call these measurements negligible. Lin et al. (2011) measure concentrations of 0.58-2.92 nM (mean 0.93 nM), which is substantially higher than Fe-limited waters, and would be classified as Fe-replete marine waters.*

We do not refer to the measured concentrations, we refer to the enrichment i.e. the Fe above background levels i.e. Considering the difference between Fe in and outside an iceberg fertilized patch. Whilst Lin et al., found low nM concentrations around an iceberg, they did not find significant enrichment looking at the distance from icebergs. One issue is likely the intermittency of the supply (as mentioned in the text 258-259), but the other being specific to the Weddell Sea that ‘background’ Fe concentrations aren’t actually that low in this region making it difficult to detect a local input (as discussed^{10,14}).

- L209-215: *There seem to be a lot of assumptions used in this calculation with relatively little discussion of uncertainty.*

We have further emphasized that this is an exploratory calculation, the only purpose it serves is showing that using existing data it can be argued that icebergs produce extremely high (unrealistically so) fertilization effects, or moderately low fertilization effects depending on exactly how Fe is scaled to primary production and primary production is scaled to C export. For clarity we have added some additional sentences to clarify this e.g. lines 412- ‘In summary, the utilization of iceberg-derived Fe in the marine environment is challenging to determine and a broad range of C export scenarios are plausible for any specific Fe addition and ice melt mixing scenario’ and to emphasize that the relative utilization of iceberg-derived Fe remains unknown (hence why we plot 1% and 100% scenarios) (lines 346-352, 276-280, 398-411).

- L214-215: *1% of what? Total flux? Median Fe scenario? Mean Fe scenario?*

For a regional-scale Fe utilization calculation based on mean iceberg Fe content which is how iceberg Fe input and its effects have been calculated to date¹⁵. (This line is removed following comments concerning the validity of the utilization approach as it makes no sense to question the validity of the approach on a regional scale-line 335- and then go on to deduce a utilization value).

- L228-231: *Isn’t this similar to most other delivery mechanisms? Usually the flux at gate is given, without identifying inefficiencies in delivery. Given this, how efficient are icebergs compared to other input sources?*

In some respects yes, for example dust loadings obviously vary spatially and temporally. Yet in models or calculations concerning the biogeochemical impact of this dust, this is usually accounted for by scaling the Fe input to the dust loading using either measurements of atmospheric deposition or some proxy for recent dust addition like Al. With icebergs the flux referred to in literature is scaled to ice volume similarly to how it would be for freshwater runoff or for rainwater but the heterogeneity is much more pronounced at source.

For rivers, for example, a 'removal factor' of about 90% is applied, but the observed removal factors actually vary between 60 and 99%. For glacier rivers for example see^{16,17}. Changing this removal factor in flux calculations has a far larger impact on the calculated flux than seasonal variation/inter-catchment variation in freshwater Fe concentrations. Both pre- and post- estuarine removal fluxes are frequently presented in the literature. A multi-Fe-source efficiency comparison is very challenging because different mechanisms (e.g. dust, rivers, icebergs) have different removal mechanisms and scale differently with environmental factors. Dust is a prime example, as alluded to already, high dust loadings can decrease Fe availability i.e. be a negative influence on dissolved Fe. We have tried to raise the key reasons for this in the text without an extensive comparison to different Fe sources which would be beyond the scope of the text (lines 299-337).

- L237-238: *This needs some additional explanation.*

Extra sentences are added to discuss observed C export, C export efficiency and how this is thought to scale with primary production ().

- L249-266: *Okay – I think this conclusion is nice, but it doesn't really back up the closing line of the abstract. The main findings that I interpret are the a) there are inefficiencies in iceberg transfer of iron to the open ocean, b) that these need to be accounted for in global biogeochemical models, and c) Fe flux does not necessarily imply C export (although I think this is the weakest area of the paper at present). I don't think the argument of little additional iceberg Fe fertilization under future warming scenarios is particularly well substantiated (especially for the Southern Ocean) at present. There is also relatively little comparison to other Fe sources (and the efficiency of those inputs), which I would like to see in order to contextualize the importance of iceberg Fe fluxes.*

The comparison to other sources, specifically focusing on short-term processes that drive divergence between the gross and net input of Fe into seawater is a useful suggestion, but we find this challenging given the timescales involved as the dominant removal mechanism varies between sources. E.g. freshwater (ground-water/rivers) supplied Fe is generally removed due to flocculation over timescales of minutes-hours, dust derived Fe is scavenged over timescales of minutes-days, shelf sediment derived Fe is oxidized and scavenged over timescales to hours to weeks; yet all of these removal processes are poorly defined quantitatively. It is not possible to simply tabulate or quantify these timescales/factors without very extensive discussion; and for icebergs the 'removal' factors, or transformation between TdFe addition and dissolved Fe in seawater are practically undefined.

We have however added some additional discussion of factors that potentially make icebergs more/or less efficient sources compared to other mechanisms. The conclusion has also been re-phrased, noting that we only demonstrate numerous changes to the nature of icebergs (apart from total calved ice volume) have the potential to affect Fe fluxes and not that they actually will do under future discharge scenarios.

- *Table 1: There's a lot of information in this table which makes it quite difficult to follow. To those less familiar with oceanography literature it would help to clarify some terms (such as mean export efficiency – i.e. fraction of C fixed from NPP exported below 100 m). Also, it is unclear to me how the mean C export*

is derived from these values. For example, export efficiency and NPP are lower for scenario (e), yet mean C export is higher (compared to (a)-(d)). Some additional explanation would be useful here.

We have simplified the table to emphasize the key point, that how Fe inputs in the literature are scaled to C export can be very misleading as ballpark figures can produce unrealistically high C export (e.g. multiplying mean iceberg Fe content by a typical C export efficiency). An additional paragraph is added with definitions and a line introducing each term in the main text.

According to the observational relationship between C export efficiency and primary production, it is possible for higher primary production to drive lower C export due to the pronounced decline in C export efficiency at high primary production¹⁸. Although this is not captured in global biogeochemical models, and widely suggested to be simply an artefact of the time differences between surface primary production and C export to depth, it has been shown using both sediment traps and thorium data (which integrates over longer time periods) and in both spring and summer data¹⁸ making it a difficult to support the argument that it is a methodological artefact and very difficult to reconcile with modelled C export efficiency.

- Figure 3: Could the authors indicate the median Iceberg-derived Fe range here (i.e. 28-960 nM as in the text)?

Of course. Added. Please note there was an error in the median Fe values originally presented as Godthabsfjord was included as two data subsets rather than as a single category, the range is now 44-960).

Reviewer #2 (Remarks to the Author):

The manuscript by Hopwood and colleagues explores the role of icebergs as sources of iron to polar oceans. The authors initially approach this topic by presenting an impressive set of field data, documenting iron concentrations of iceberg samples collected from a range of systems. Using this observational dataset in conjunction with modeling of iceberg iron release as a result of melt and carbon export resulting from iron addition, the authors then evaluate large-scale contributions of iceberg-derived iron for carbon export in the Southern Ocean.

The paper's topic is of great general and personal interest, and its submission timely, given the numerous recent results in the physical literature concerning the importance of iceberg and iceberg melt for freshwater fluxes (e.g. Moon et al. 2018, N. Geoscience), and the growing body of literature demonstrating ecosystem consequences of ice-ocean interactions. To this reviewer's knowledge this is the most comprehensive survey of iron concentration in icebergs to date, and this aspect of the research itself makes it quite novel. The combination of observational data with modeling of iceberg decay is also an interesting approach, particularly given the extensive supporting data provided by the physical observation collected in Sermilik Fjord.

While I find the observational dataset quite impressive, and the questions being addressed of great worth and general interest, I have qualms with the ultimate scope of the paper, and the approach taken by the authors to link their observational dataset to Southern Ocean carbon fluxes derived from iceberg melt.

1) Whether icebergs can serve as a significant source of iron to the ocean and primary producers is a central question of the paper. Instead of addressing this question, the authors take a particular stance on the issue early on in the paper.

This is perhaps a stylistic point, we of course wrote the paper after we had finished analyzing the samples/data. Given the comments on the limitations of the utilization approach, which relies on using satellite derived chl a to deduce Fe demand, we have removed much of this text and introduced the topic more through highlighting the large discrepancy in iceberg fertilization efficiency in different models.

A central assumption of the paper, presented in L85-95, is that there is inefficiency in the delivery of iceberg derived iron to primary producers. The authors cite a paper by Boyd and colleagues as supporting evidence for this assumption (specifically, in the case of iceberg supply, values derived from Shaw et al. 2011 for Iceberg Alley), which they state show far greater supply of iceberg iron than biological utilization. Estimates of biological demand / utilizations are in this reference based on a combination of lab and field-derived Fe:C ratios, as well as remote sensing datasets of primary production. Hopwood and colleagues present this mismatch as a fact, which is misleading considering the context in which this data is presented in the original paper. While acknowledging the gap between their estimates of iceberg iron supply and utilization, Boyd et al. make a number of qualifying remarks:

"The rates of iron utilization appear to be considerably less than that potentially supplied from iceberg melt (Figure 6d c.f. Table 2) suggesting some temporal mismatches between supply and utilization. However, determining the extent to which Fe utilization in this region is met by icebergs is not possible due to the confounding effects of overlap with other Fe supply mechanisms such as Patagonian dust and resuspension of shallow sediments (Figure 6)."

This is an interesting comment, overlap of ample Fe sources is itself often an inefficiency worth mentioning. This is clarified in the introduction as overlap alone, as noted, is another potential reason for inefficiency in any Fe source – not just icebergs- particularly in coastal areas where many source do overlap. In any case, the main point which we failed to stress concerning the use of the utilization ‘ballpark’ calculation in Boyd et al., was that it used an old and very broad estimate of mean iceberg-Fe content. Updating this with our value makes the total Fe input higher. This would increase the gap between utilization and input. However, another key point is that we don’t think such an approach is valid; integrating a total Fe input over a large range is possible only for Fe sources which are roughly ‘evenly’ spread, and the point of this manuscript is that the total iceberg Fe flux used to date is too high for oceanographic calculations as it refers to the total Fe input into the whole ocean, and not the bioavailable Fe input into the surface mixed layer which remains challenging to calculate, but is certainly much lower.

or otherwise, concerning the dataset as a whole:

"The circumpolar estimates of phytoplankton Fe utilization that we produced cannot readily be compared with that for Fe supply, since the latter estimate will be much less spatially resolved (Figure 5). Furthermore, major uncertainties exist regarding the geographical realm of influence of each supply term and in some cases, little is known about putative mechanisms such as Southern Ocean eddies (Table 2). Nevertheless, our initial comparisons between the magnitude of Fe utilization and supply revealed that they are of the same order, and specifically that both regional match-ups and comparisons of maps of regionally distinct Fe utilization with the Fe supply revealed no major mismatches between the magnitude of utilization and supply."

Boyd et al. further acknowledge the large uncertainties in their estimates of biological utilization, stating for example that:

"The sensitivity of Fe utilization estimates to altering either Fe:C ratio or the threshold Fe concentration [...] is greater for the former (Figure 12), with changes in the magnitude of Fe utilization increasing, relative to the standard run, by ~twofold, depending on what set of Fe:C ratios was employed."

While the cited paper therefore claims no statistically significant mismatch considering uncertainties, Hopwood et al. take this difference as a given, basing the rest of their analysis on this premise.

It was our understanding that the 'circumpolar estimates' refers to the overall Fe supply vs utilization relationship and not specifically to icebergs. The observation of total supply and total utilization matching on regional scales is not inconsistent with the earlier comment that one source may not match in one specific region. Note also that the Boyd et al work uses a very broad range of iceberg Fe content (over an order of magnitude), derived from the limited measurements and some indirect estimates available at the time. A key point of this work is that we verify iceberg Fe concentrations are at the upper-end (or even larger than) this early work, so the utilization as calculated using 'old' Fe estimates declines if our larger dataset of direct measurements is applied. Our analysis is not based on the premise that iceberg Fe fertilization is inefficient. Our analysis is based on a dataset which shows that Fe is highly heterogeneous in icebergs on a global scale and exploring the implications of this.

It is also worth noting that iceberg affected regions considered by Boyd et al. and in this paper are limited to Iceberg Alley / Weddell Sea, based on remote sensing dataset (sourced from BYU) tracking large (> 5 km) icebergs visible via satellite. Examination of spatial maps of iron utilization in Boyd et al. points to large heterogeneity in biological utilization, with peaks in the West Antarctic Peninsula, Southern Weddell Sea, Amundsen Sea, and Ross Sea, in polynyas or otherwise at the margins of glaciers and ice shelves where icebergs (potentially smaller) would be expected to calve. These sources of icebergs/iron are not considered in Boyd et al. or in this publication (though observations are all collected from smaller systems outside Iceberg Alley), although icebergs are routinely seen in all of these regions.

Yes, there are several additional Fe sources in these regions which overlap (as now noted in the text), but budgets from these regions do not presently consider iceberg-derived Fe to be a large source-term. The 'pumping' of Fe from sediments due to subsurface melt release in such environments was not specifically known at the time of the work in question and this is now thought to be a significant Fe supply term in these environments which is closely linked to shelf sediment derived Fe supply^{19,20}. This doesn't really affect any of our results, although it is worth, again, highlighting that the coastal ocean is a place where multiple Fe sources overlap and thus determining the significance of Fe delivery from any one source into coastal waters is always challenging.

In general, model based estimates of utilization and availability should be interpreted with caution. For example, the amount of soluble iron derived from aeolian dust the authors quote are, to this reviewer's knowledge, chosen to fit surface ocean iron observations in a modeling / large scale framework, yielding values of 1-10% solubility (an enormous range). Some of this nominal solubility may include abiotic scavenging in the marine environment.

The model components underpinning the estimates of atmospheric Fe deposition within Boyd et al are subject to less, or comparable, uncertainty to any other Fe source. Yes 1-10% is a large range, but not compared to the uncertainty on iceberg derived Fe, for example, at the time this earlier manuscript was written. Furthermore the variation in atmospheric Fe supply-completely beyond the

scope of this paper- is at least partially explained by changes in loading, origin and other biogeochemical factors- whereas the variation in total iceberg Fe supply was/is simply because we didn't have any data to quantify it.

The bioavailability of glacier derived iron is a very important topic and one which is deserving of greater attention, via, for example, incubation experiments or seawater additions exploring how much enters the dissolved phase, and how much is taken up in organic biomass. This is beyond the scope of this paper. However, I find the premise of a mismatch a great overreach given context presented in the cited paper, and the use of models without careful discussion of uncertainty, error propagation, or corroborating observations dissatisfying.

It is unclear what model the reviewer is referring to here. The iceberg-melt 'model' is extensively corroborated for the environment in which it is used here with extensive CTD profiles²¹. Hence why we chose it, iceberg-melt within Sermilik fjord is extensively studied and all melt rates herein are derived from the most extensive observations of icebergs available for any catchment globally. There are no other model results in the manuscript.

We now specifically raise the 'unknown' of how TdFe enters the dissolved phase, as this is of direct relevance to the potentially biological impact of icebergs, but something that cannot be directly defined from particulate data. Additionally, we have re-written the later part of the manuscript to raise explicitly questions which cannot yet be answered by existing data.

2) In examining contributions of icebergs to the environment (i.e. iron concentrations and carbon export), the authors treat icebergs as passive sources of iron, delivering meltwater into the water column without considering the oceanographic implications of this freshwater transport. In this sense their approach is akin to Hawkings et al. (2014), in which a time series of iron concentrations from a proglacial stream is used to derive a flux of iron to the marginal ocean. I emphasize that in both cases the observational datasets underlying the studies are of tremendous significance and importance. However, overarching conclusions concerning impacts on the ocean drawn from such observations should be treated with caution.

In contrast to Hawkings et al., the data in this paper is used to make the opposite point, namely that the delivery of iron from glacier is inefficient, with numerous reasons for this inefficiency. Several papers, include Hopwood et al. 2018, have considered the impact of freshwater delivery on coastal ocean dynamics, demonstrating that freshwater input serves to impact vertical circulation, with consequences for nutrient fluxes and distributions. A paper cited by the authors in this paper, Lin et al. (2011), also notes that subsurface iron concentrations at the margins of icebergs was elevated, which they attribute to upwelling of iron-rich circumpolar deep water driven by basal melting.

Boyd et al. speak of "confounding effects of overlap with other Fe supply mechanisms," which would include upwelling at iceberg margins, or resuspension of sediments both as a result of advection from continental shelves where icebergs are calved but also iceberg grounding along the continental shelf, evidence for which is presented in the geological literature and is particularly relevant for large tabular icebergs in Iceberg Alley.

Icebergs shouldn't simply be considered as passive sources of freshwater to the marine environment, if sweeping claims as to their contributions (or lack thereof) to polar ocean productivity and carbon export, and the Southern Ocean in particular, are made. Interactions between icebergs and oceans should be

considered. However, Hopwood et al. make no mention of feedbacks between melt and circulation. At a minimum this source of uncertainty should be acknowledged.

In fact, an interesting development of this study would be to use the melt model discussed in L121-169 to explore vertical nutrient fluxes at iceberg margins associated with ocean melting of iceberg, given the extensive physical data for Sermilik fjord, an important freshwater export pathway for the Greenland Ice Sheet. Hypotheses could be generated, and measurements to test these hypotheses could be collected.

Upwelling from icebergs is very minor compared to that from glaciers and it could be misleading to compare the two. Glaciers generate buoyant plumes which rapidly rise, typically to the surface before reaching neutral buoyancy at the surface or at depth. These are in the case of the glaciers referred to in the above studies—principally driven by subglacial discharge. If we calculate the relative importance of upwelling of macronutrients around glaciers compared to the direct freshwater input, the upwelling effect indeed dominates (largely because freshwater doesn't contain much macronutrients) producing >90% of the surface/near-surface macronutrient enrichment downstream for NO_3 .^{22–24} But with Fe, the opposite is the case (for an idealized glacier), the freshwater source always dominates over upwelling. Putting some best guesses of ambient Fe data into the formulation used for NO_3 upwelling in Hopwood et al 2018, would suggest that between 50–99% of the dissolved Fe downstream of a glacier comes from meltwater and 1–50% from upwelling. (We recently did this exact calculation out of interest in a review text, but noted that we simply cannot reduce the uncertainty yet using available data²⁵). The uncertainty is high because of difficulty in constraining how dissolved Fe behaves over the glacier-to-ocean salinity gradient. Beyond the complication of dissolved Fe behaving non-conservatively, there is the added complication of sedimentary Fe from also being 'pumped' within buoyant subglacial discharge plumes^{19,20} and the possibility that dissolved Fe becomes saturated during the mixing process between glacier- and marine- endmembers^{26,27}. Unlike macronutrients, we do not have extensive near-terminus Fe speciation profiles to reduce this uncertainty.

In any case, moving to icebergs, the upwelling is much weaker, does not systematically reach the surface/near-surface waters and the uncertainty in the fractional importance of dissolved Fe from upwelling and freshwater is even larger. Subsurface melt is also poorly parametrized²⁸ and so a plume calculation for icebergs with respect to Fe inputs would presently be a very difficult exercise. Many biogeochemical papers discuss, briefly, the prospect of 'upwelling' around icebergs, but there is little we can presently do to model the effect of this on Fe due to a paucity of data and a lack of well-developed physical models for iceberg melt to incorporate biogeochemical data into. Melt on iceberg faces is very weak compared to glaciers where subglacial discharge is actively released at the grounding line. 'Upwelling' downstream of marine terminating glaciers is well-supported by field evidence, but these systems have a large meltwater perturbation which is 'easy' to trace, and can be approximately parametrized as static ice bodies with a 2D circulation leading up to the ice edge. Icebergs are, as noted, conceptually much more complicated, cannot be considered static, and the melt is a far weaker signal in the water column making it difficult to generalize between different icebergs at different times subject to slightly different water column conditions^{13,29,30}. We recognize this is an important issue to raise, and have brought the physical fate of ice melt into the discussion of uncertainties with one idealized model presented in supplementary material just to illustrate the point. For a mean Fe input from ice, freshwater would always dominate the input relative to upwelling almost irrespectively of the mixing scenario simply because the mean ice Fe endmember is high relative to seawater. But with a realistic range of iceberg endmembers, any scenario (with massive uncertainty on the total flux) becomes plausible: the upwelling supply could dominate at low

freshwater Fe concentrations, whereas the direct freshwater input is 100% of the Fe flux at high freshwater Fe concentrations.

3) The paper progresses from discrete observations collected from a range of systems and icebergs across the globe, to a discussion of iceberg melt and iron loss based on a particular Greenland fjord system (Helheim / Sermilik), to an analysis of carbon export in the Southern Ocean. In aggregate these vignettes are designed to support the premise of inefficiencies in the contributions of iceberg to ocean carbon export and iron delivery. The extent to which conclusions from one section relate to the next is however unclear in my mind. For example, hydrography in Helheim / Sermilik shows a warm (~ 4 °C) water mass at depth with a cooler (~ -1 °C) water mass above. This water mass distribution is not representative of what is found in the Southern Ocean, where the warmest water mass, Circumpolar Deep Water, reaches a maximum of ~2 C offshore, and significant decreases in temperature are noted across continental shelves to the glacial margins. Circulation patterns in coastal bay / fjord systems are also quite different in Greenland v. Antarctica, with currents in Sermilik reaching tens of cm/s, while such velocities (along with temperature a major determinant of melt rates) are generally not found in coastal fjords of the Antarctic Peninsula, for example. Water masses found along the Western Weddell Sea shelf are generally far colder than along the West Antarctic Peninsula, with icebergs frequently grounded or otherwise circulated along with sea ice. All these factors would impact melt rates, iron loss, and thereby spatial extent of iron inputs, although they are at best glossed over in this article.

This is explicitly stated in the text, as is the justification for Sermilik and it is clearly acknowledged that Sermilik is a relatively ‘fast melting’ scenario. Sermilik was chosen by necessity because it is (to date in the literature) the sole catchment where any such calculations can be performed. We add some extra lines in the text to raise the issue again of inter-catchment differences. Further, in Supplementary Material, we outline how different water column properties affect the fate of melt (and thus Fe) in different water columns contrasting a relative ‘fast’ ice melt region (S Greenland) with a relatively ‘slow’ melt region (Antarctic Peninsular). But we note that further work to combine physical parametrizations of melt with nutrients (especially Fe) and ice-ocean relative velocities is beyond what can be presently substantiated with available data.

Considering an iceberg as a system, its impact will inevitably depend on its age, as the authors discuss, as well as the distribution of iron within icebergs, which remains unknown given that only marginal iceberg samples have ever been collected. However, it should also depend on what particular system the iceberg traverses.

Yes, we have added a brief comment to the effect that the effect of icebergs of course varies with where the iceberg is.

The authors largely discount "hotspots" of productivity as contributors to carbon export (L258) based on a parametrization of carbon export, despite observations of the importance of such systems from process studies, and of episodic pulses of production, for carbon export and marine ecosystem processes. Gerringa et al. (2012), and Annett et al. (2017), as well as references therein and citing papers, speak of episodic iron limitation along coastal Antarctica, as well as rapid utilization of glacially derived iron by primary producers.

There is only one process study (we are aware of) that presents data to suggest icebergs may have a more efficient export of C relative to background export, which we explicitly refer to³¹. This study³² compares C export in, and out, of an iceberg fertilized area concluding that C export was higher following iceberg fertilization. It does not compare C export efficiency in the iceberg affected region to

data elsewhere in the S Ocean, nor does it demonstrate that icebergs are ‘hotspots’ of C export efficiency relative to the S Ocean. If we investigate this (using the data for which we can find a correspond primary production^{31,33}), it is not apparent that iceberg associated C export efficiency is high compared to that we would expect anywhere else in the Southern Ocean (see below).

Therefore we disagree with the reviewer’s specific comment (underlined above) and suggest that the findings of Shaw et al.,³² have been in some cases miss-cited as implying that icebergs promote high C export efficiency relative to that observed elsewhere in the Southern Ocean- which the data presented does not support. Icebergs may well promote hotspots of primary production relative to background waters locally, and may increase C export relative to background waters locally. But this is not the same thing as icebergs acting as hotspots for productivity and C export efficiency relative to the general patterns in productivity and C export across the S Ocean (which is what several manuscripts have used this specific reference³¹ to imply).

Figure: Thorium based C flux estimates and corresponding estimates of primary production from an iceberg fertilized region (red squares)^{31,33}, and from the Southern Ocean dataset comparing multiple methods/seasons assembled by Maiti et al., (blue diamonds)¹⁸.

The other references in the comment above refer to surface primary production, not C export. There is no doubt expressed herein that Fe from any source into the Southern Ocean can have a rapid positive effect upon primary production. We acknowledge that C export could be systematically enhanced by the presence of icebergs locally, but this is –in literature on the subject– suggested to be a local phenomenon due to the community composition within a few km of icebergs³³ and not a broad-scale phenomena. Furthermore, this hypothesis refers to C export, not C export efficiency. C export efficiency could also be locally enhanced by icebergs, but we cannot find any evidence to directly support this hypothesis and the limited values available for icebergs are not unusually high (Figure). Furthermore, the question addressed here, and within global models, is whether icebergs have a broad-scale fertilizing effect. If they do, then it is not plausible that the resulting broad-scale fertilization from icebergs is both large² and has a C export efficiency relationship to primary production far higher than that either observed or modelled in the Southern Ocean⁷.

Examining these process from a mean / steady state perspective, using even regionally tuned (remote sensing-based) models, likely hides variability which would have significant impact on the conclusions presented in this study. The topic of carbon export, and its parametrization from large scale (model, remote sensing) observations is the subject of significant ongoing study, including the NASA EXPORTS project. At a minimum, caveats should be presented regarding uncertainties in the model used to derive primary conclusions, and mismatches between export calculations from large-scale datasets v. in situ traps.

We do not use any modelled POC results, we use Southern Ocean observational data. This dataset, the most comprehensive collected, includes multiple seasons and 2 distinct methods (traps and thorium) meaning that the observed trend (which doesn't vary between the methods) is reasonably robust and cannot easily be dismissed as due to mismatches in the timing of surface primary production and sub-surface C export collected by sediment traps¹⁸. We are aware that this is subject to ongoing debate, and is particularly negatively received in the modelling community because it suggests that the temperature-dependent curves used to model C export efficiency are insufficient to explain observations, but also note the multiple methods used to demonstrate the decoupling between surface primary production and C export to depth³⁴.

There may well be variability on smaller scales in iceberg fertilization concerning the local effects- in fact these are practically inevitable due to the dilute nature of iceberg melt and the heterogeneous distribution of Fe¹³, which we can of course acknowledge, but we clearly state in the text that we are discussing the broad-scale effect of icebergs.

We now explicitly discuss the prospect that icebergs depart from regional patterns by having their own primary production – POC export relationships, but we note that this is not the 'fertilization' that global models are concerned with i.e. any peculiarities of the biological community around icebergs³³, or the sediment release beneath them etc^{31,35}, may well drive unique biogeochemical features. But this would not in any way explain any of the differences presently observed using broad-scale chl a data^{7,8} or modelled iceberg impacts^{1,2} to estimate the effects of icebergs on the C cycle. These entirely arise from either the assumed Fe input, or how that Fe input is computed to produce a POC output, or how surface chl a data is scaled to POC export.

I reiterate that this paper presents very interesting data that contributes significantly to our understanding of iron concentration present in glacial ice. The paper also presents intriguing model results concerning the contributions of iron to the marine environment, and timescales over which this iron is lost, which provides insight into the spatial extent of its potential impact and results in testable hypotheses. Their research, as summarized in L226-228, provides valuable insight into processes that may significantly impact iceberg contributions to the ocean. However, I find the paper, in aggregate, overreaches in its interpretation of models given the observational data, in order to fit the narrative that icebergs are an inefficient source of iron, which is at best a conjecture and not as this paper presents a given. In this respect their concluding paragraph (L258-266) is telling, as they urge better parametrizations for models based on their work, but make no mention of the need for field experiments and observations (process studies, incubation experiments) to test their hypotheses regarding iceberg iron distribution, bioavailability, timescales of iron loss from icebergs, or carbon export resulting specifically from glacial iron input.

A key point, not written in the manuscript explicitly, is that because of the heterogeneity in ice, more observational field studies alone simply trying to trace Fe around icebergs and its effects would be of

debatable use because the input is so heterogeneous that when the added complications of iceberg movement are added, field studies would have extreme difficulties in even defining exactly what region was directly affected by iceberg melt and in tracing the iceberg-derived Fe (which is consistent with the extensive comments already written concerning this subject)^{10,14,36}. That being said, there are of course key challenges that only future fieldwork will overcome, for example defining the fraction and nature of ice present in basal layers, and defining how ice melt plumes behave in the dynamic surface ocean. We now discuss these required developments in the context of reducing uncertainty.

There are a number of interesting ways to develop this study, including a broader discussion of under what conditions, for a range of systems (i.e. the ones that have been sampled), icebergs may or may not be an important source of iron, or using a melt model as presented in the second section whether ocean sources of iron, sourced from upwelling, are significant compared to what is exported from the iceberg itself for the particular system where data was collected.

We suggest that this is not compatible with the reviewer's own comments, there is practically no field data (for obvious reasons given the difficulty of deploying CTDs immediately next to icebergs) with which to develop such models. An upwelling model for macronutrients close to icebergs may be possible, but is subject to large uncertainties, for Fe even an upwelling model (in 2D) for a marine-terminating glacier has proven challenging to develop, a 3D upwelling model for Fe based on practically no field data is presently an almost pointless exercise because the uncertainty is massive relative to any calculated flux. In any case, the upwelling effect of icebergs certainly does effect the distribution of any biogeochemical parameters in the water column, but considering the relative size of the freshwater and saline nutrient pools, is likely very weak for Fe compared to macronutrients (see above comment) under most circumstances.

Further work is certainly needed to parametrize (using field data) the nature of Fe-rich layers within ice, and to better parametrize subsurface melt²⁸, and melt plumes in a dynamic water column, before any attempt can be made to properly couple ice melt models with water column Fe concentrations/distributions.

One could also explore timescales of iceberg transit and melt in the Southern Ocean from available datasets, with model parametrizations provided by large-scale models (e.g. SOSE). This datasets will make a significant contribution to the field. Simplifying this manuscript to focus on a particular aspect of the story and hypotheses, while giving a broader perspective on the question of iceberg contributions, will make for a stronger discussion.

There is already extensive satellite work tracking icebergs in the literature which, we note, again disagree on the magnitude of the biological effects⁷⁻⁹. We are not sure that this suggestion would help simplify the manuscript, especially given the large uncertainty that would arise with the vertical distribution of melt within the water column (as now discussed in the raised text) and the uncertainties with use of chl a data at high latitudes (critiqued in earlier review comments).

Minor comments:

L46-48: The sentence discusses fluxes, but a concentration is presented.

We referred to the concentration used to derived the flux (now clarified).

L59-61: additional details as to this analysis need to be presented in the methods. for example, a multiple comparison test was presumably run to derive this result, although this is not detailed anywhere. was the ANOVA run on log-transformed data?

No, ANOVA was run on all concentrations grouped by region and by catchment. Now tabulated in supplementary material.

L96-98: If icebergs are characterized by extremes in concentrations, with regional variability, and time variation in concentration, a mean concentration is likely an uninformative quantity, particularly given the skewed nature of the distribution of iceberg iron concentrations (e.g., as seen in the distribution of log10 concentrations presented in Figure 1).

Indeed, this is basically the key point of the paper. Yet it is the mean that has been used previously to derived the iceberg Fe endmember and to inform models. Hence why we raise the mean for comparative purposes, but go on to stress that the median is more meaningful and even a point-Fe source defined around the median still likely unsatisfactory for approximating an iceberg-Fe input in the ocean.

It's worth remembering that the iceberg samples under consideration, while (again) of tremendous significance, represent only a small subset of all icebergs, that samples were collected only from the outer portion of any given iceberg, and that large tabular icebergs characteristic of Antarctica / Iceberg Alley were not sampled, based on the iceberg dimensions and locations presented in the manuscript.

Of course. This itself may induce bias in the results if the fractional importance of basal ice changes regionally. But this is completely unknown with respect to 'real' icebergs in the ocean. We add some discussion around this topic.

L193-194: this is in contradiction to the results of the ANOVA discussed in L59-61

No, only the Icelandic samples were found to be different, we clarify that these were then excluded from any further calculations.

L271-273: It's unclear from the text how the described method yields randomized data collection, given that this sentence speaks of "regular intervals" and "predefined transects". Were transect starting points and headings randomized? how long were transects? Were any icebergs omitted from sampling due to safety concerns or other?

We are not sure this matters. Headings of course were not randomized because we had to sample within ice fields, boat drivers were instructed to cruise along agreed transects through dense-ice areas and then a sample was collected, processed (5-15 minutes work to get ice on deck and transfer it to a lab or clean area without contaminating it) and then the next sample collected. Transects varied depending on location and lasted from a few hours to a day. The purpose to this was to avoid samplers 'cherry-picking' samples that looked particularly interesting e.g. prior work has focused on collecting sediment rich icebergs i.e. driving around until locating samples that look 'dirty' and chiseling out the basal/sediment rich layers to collect a sample. We would rather not provide guidance concerning safety in the text as proceeding close to icebergs is obviously an inherently dangerous activity. In these catchments no icebergs were specifically avoided.

Reviewer #3 (Remarks to the Author):

This work by Hopwood et al is novel and extremely timely. Many researchers speculate that the predicted increase in the frequency of iceberg calving events will result in enhance Carbon export, especially in the Southern Ocean. This is because the Southern Ocean is generally anemic and that icebergs have high iron content. Icebergs can therefore become "hotspots" of biological activity. The paper claims that the general view that iron is distributed homogeneously within icebergs has led to skewed estimates of icebergs fertilisation potential, and therefore carbon sequestration.

I find particularly outstanding the wealth of observational data collected in this study. 210 icebergs samples, spread across both poles, is unprecedented. Results are clear, showing extremely large gradients (6 orders of magnitude) in iron content. The field data is then complemented with a model to evaluate the effect of iceberg melt waters, considering 3 different iron distribution scenarios. The age of

the icebergs appears to strongly influence the fertilisation potential of icebergs, which makes sense, but has not explicitly been put forward nor studied before. The study does on to then evaluate carbon sequestration under different scenarios.

The paper is beautifully written. The introduction is clear and the methods are adequate. Again, collecting pieces of icebergs in the most remote and challenging parts of the world ocean, under trace metal clean conditions, is commendable. I find the section on the inefficiency in iceberg-iron delivery especially convincing. It gives a clear path to why this work is needed. The main findings are very well supported by the field and model results. This paper will undoubtedly be highly cited in the field of chemical and biological oceanography, especially for researchers working in polar waters.

The reviewer is thanked for comments on the text.

Reviewer #4 (Remarks to the Author):

Review of "Highly variable iron content limits iceberg-ocean fertilisation and carbon export" by Dr Hopwood and co-workers

This is my first review of the manuscript, and it will focus particularly on the iceberg modelling approach. The authors present a novel collection of more than 200 global iceberg samples, which shows that iron concentrations within icebergs vary over six orders of magnitude. Using an iceberg melt model applied to an Arctic fjord during summertime, the authors proceed to show that the spatial variability/heterogeneous spatial distribution of iron within icebergs (e.g. concentrated in the basal layer, or as a shell layer surrounding the iceberg) would lead to a rather rapid loss of iron within a few weeks of calving, thus limiting the delivery of iron to the more remote areas of the Southern Ocean, where iron-limitation tends to be stronger than close to the continent. This will therefore also likely limit carbon export. The rapid loss of iron within a few weeks after calving appears to be supported by an analysis (in a previous study) of ice-rafted debris in a sediment core from an Arctic fjord.

The findings of the authors are novel and the fertilization effect of icebergs and the associated potential negative climate feedback are still poorly constrained and under debate. The study by Dr Hopwood and co-workers is therefore timely and of large interest, as the authors also conclude that climate-driven future changes of the iceberg meltwater pattern might impact C export to a larger degree than the potential increases in the total iceberg iron flux itself (which is expected to increase under global warming). This highlights new model development directions for the climate modelling community. The paper is written clearly and the quality of the figures is very good. I therefore recommend minor revisions as detailed below.

Thomas Rackow

1) One concern is that "the fraction of iceberg-derived Fe that can be transported to the open ocean" (l.50/51) has been constrained using conditions within a single Arctic fjord during summer, and these results are further used to quantify fertilization and C export "in the Southern Ocean" (l.52). This is stated once more in lines 231-234. I am aware of the data situation, and the authors state that "reduced uncertainty concerning [...] Fe loss in other catchments will only be possible once further observational data becomes available", but the paper would benefit from a short discussion why these Arctic results are potentially transferable and applicable to the Southern Ocean conditions.

The conditions in any one-fjord are unlikely transferrable to any other fjord/glacier as many properties of coastal waters can, and do, affect ice melt rates and residence time in specific regions.

We acknowledged in the original text that Sermilik is anticipated to be a catchment with a fast melt relative to the global situation. The point of the scenario was to demonstrate that ice-Fe-distribution matters and to attempt to define quantitatively on what timescales it matters. We are only aware of one other catchment in which similar calculations can be performed using sediment load as a close proxy for Fe load (also Greenlandic and similar conclusions were reached compared to our work⁴). Moving from Greenland to Antarctica, the general principles of iceberg-Fe distribution appear to be similar (because our data suggests no critical differences between geographical regions – with the exception of Iceland), so the fundamental finding that Fe is heterogeneous, and a mean Fe input is a poor description of icebergs melting in a 4D ocean holds (icebergs are presently treated as a point source of Fe in the surface ocean –although some recent work has begun moving to more complex sub-surface parametrizations¹).

That being said, there are of course unknowns, and a key area for future research is how basal layers interact with the ocean subject to different water column properties which we now outline.

- l. 170ff: It wasn't clear to me how exactly the Arctic model results (in Fig.2) enter the discussion for the Southern Ocean case (in Table 1 and Fig.3)?

All iceberg-Fe work to date has principally used data from the Arctic to define iceberg-Fe content. In this work we showed that it doesn't, apparently, matter in the sense that a global mean is similar to an Arctic or Antarctic means, and in both cases the distribution of Fe is similarly biased towards a small fraction of ice (a few %) containing the vast majority of Fe. The only use of Arctic data in this context is within the global mean (in Table 1 and Figure 3). However we also speculate that there may in fact be fundamental differences in the mechanisms via which Fe is incorporated into ice contrasting small marine-terminating glaciers with large ice shelves, but these are not possible to test with existing data.

- l.174: "4% of the ice contains 91% of the Fe"; This is true for the global dataset. Because of potentially different mechanisms/iron sources in the Northern and Southern Hemisphere, do you get different answers if you do this for Greenland (Grn) and Antarctic (Ant) icebergs (Maxwell Bay, South Bay, maybe Pia fjord) separately? In particular, Fig.1c would be very interesting to see for the "Ant" case only.

Yes, we had looked at this but neglected to clearly show in the manuscript a comparison. It is apparently very similar everywhere (with, or without, Iceland being excluded). Figure 1 is amended to show a cumulative distribution defined in different ways. In all cases, it is similarly skewed.

- Fig.2 c/d: Regarding the "shell-layer" case, does this case translate to the Antarctic at all? I have seen photos of small "dirty" Arctic icebergs covered by some type of shell layer, but I am not aware of larger Antarctic equivalents of this type. My understanding is that a large percentage of iceberg mass in the Southern Ocean calves as giant icebergs from a handful of large ice shelves around Antarctica, and they might distribute iron/sediment which the ice gathered (at its base) on its route to the coast or by scraping material off the continental shelf ("basal case"). Iron-rich "narrow bands bisecting icebergs" (l.132) have also been observed (not considered here as a separate case), i.e. from atmospheric deposition. A major number of rather pristine smaller icebergs is also generated by disintegrating giant tabular icebergs in the open Southern Ocean.

The shell cases was more of an added addition to see if it matters where the surface ice is enriched with Fe, i.e. any difference between the shell and basal scenarios would be informative because (as discussed) what was originally basal ice can end up in a variety of orientations. In the context of real

icebergs, we are not aware of any literature explicitly quantifying how sediment is distributed globally, but our field experience is that 'dirty' ice resembling something akin to a real-life shell scenario occurs in shallow-fjord catchments where ice rolls in glacial flour during/after calving. Due to the relative difference in ice-coverage between Arctic and Antarctic catchments, we expect that yes this only really occurs on a large scale in the present day Arctic- although we do have field reports of such 'dirty' ice in some Antarctic coastal bays.

Yes iceberg disintegration is also an issue we didn't address herein. We have added lines to address this uncertainty discussing the fate of basal ice particularly with the updated wave-melt formulation as this becomes an important.

- l. 158/159: *"Further complexities certainly arise between catchments due to differences in ocean temperature, iceberg size distribution, and coastal ocean dynamics" This might be a good place to discuss the many differences when moving to the Southern Ocean in particular. Similar for lines 165-168. As an example, basal melting of giant icebergs might only significantly increase as soon as icebergs leave the coastal current and cross the southern ACC front (Rackow et al. 2017), which could imply a less rapid iron loss (assuming the 'basal case') and higher iron delivery to more remote open-ocean areas.*

Yes, added as suggested here and above.

2) Another minor concern is the description of the iceberg melt model itself, which as I understand has been used in a previous high-profile study, but nevertheless I'd recommend to provide some more details on the model and on the underlying assumptions for better reproducibility of the results. The authors refer to the Methods section and to the paper by Moon et al. 2019 ("Subsurface iceberg melt key to Greenland fjord freshwater budget"), which features an extensive Supplementary Information with more information about the iceberg model.

Although it appears to me that the standard melt parameterizations, known from large-scale modelling studies with much simpler idealized iceberg geometries (Bigg et al. 1997, Gladstone et al. 2001, Silva et al. 2006, Rackow et al. 2017, Stern et al. 2016, Merino et al. 2016, ...), are implemented in your model, I'd appreciate if the basic details/equations of the iceberg model would be given in a Supplement or in the Methods section. Any differences to the model used in Moon et al. (2019) should be explicitly mentioned as well.

More model details are added in supplementary material and full model code is now provided with a readme- we are very open to further suggestions if more notes are requested on this. Differences, which are minor, were already explicitly stated (now lines 482-494). We have expanded these slightly to clarify that definition of iceberg side melt can be expressed in two ways (next comment). Given the excellent notes available on the original model (which is subject to only very minor changes with respect to the physical -i.e. non-Fe- aspects, we prefer to refer to the Moon et al., text rather than presenting model equations herein.

- In previous iceberg modelling studies mentioned above, the wave erosion melt rate has often been applied to the whole side area of an iceberg; from the Supplementary Information to Moon et al. (2019), here it is only applied to a narrow strip along the waterline. Why is that? My understanding is that the parameterization accounts for the calving of overhanging slabs as well, and that's why the whole side area is usually used. Moreover, erosion is often the largest decay term in iceberg melt studies, so the model results could be rather sensitive to the choice of the side area.

We are aware this is an on-going discussion in the field and it was unclear to us which formulation was better. We previously opted for the narrow strip along the waterline approach because it minimizes the difference between the 3 scenarios in figure 1, whereas alternative approaches amplify it (along with total Fe loss). In prior work, this was clearly justified as over short-time scales (8 days²¹), the difference between these formulations is minimal and was included in the uncertainty for specific calculations²¹. However, over longer time periods comparable to fjord residence time (i.e. 80 days), the differences become larger. As we cannot find a clear justification for either approach in the literature, we therefore show both extreme cases. Melt is applied either only to the waterline, or to the entire faces. In the later case, a significant number of icebergs become unstable, as we have no extensively tested way of modeling/accounting for this and it obviously may trigger substantial re-distribution of any Fe, we simply note the point at which iceberg instability is reached (dots on Figure 1 a/b).

- Following Nature's recommendations to authors

(<https://www.nature.com/authors/policies/availability.html#code>), it would also be a good idea to share the iceberg melting code on a public repository. Given the structural uncertainty of current iceberg models, which is also mentioned in the Supplement to Moon et al. (2019), a tagged version of the code that was used in your study would greatly facilitate future research that wants to build upon your efforts.

Now added.

=====
Line-by-line comments:

l.23: "Using a global analysis of iceberg samples" It was not clear to me whether this is a compilation of previously existing data or whether a large part of the collection has been newly done for this study

In total we have collectively collected and analyzed 201 samples globally. Some of these (n = 49) have been included in prior site-specific publications for other purposes and an additional small number of datapoints (n =5) are available from older literature. This should now be clear in the supplementary dataset as the origin of all samples is indicated. This is also clarified in a few lines at the end of the methods section.

l.36/37: "and has yet to be explicitly parameterized in global ocean biogeochemical models" One could go as far to say that it has yet to be explicitly _simulated_ in models, e.g. by combining biogeochemical models with Lagrangian models of iceberg drift/decay (as in the references above). (similarly l. 259)

Yes, agreed, worded accordingly.

l. 117/118: It appears to me from these two lines that there is a known compositional difference between iron-rich basal ice and non-basal ice of lower iron content. Would you therefore consider the "basal layer" case (l.134 and Fig.2) to be the default, most realistic case of the three?

Yes, as discussed above, the closest to a 'real' iceberg is the basal case, or something similar to a basal case but with Fe-rich layer(s) randomly orientated such that the behavior is part-way between the basal and Monte Carlo scenario.

l. 133: What does "endmember" refer to? Delete?

Because the scenarios are not realistic in a biogeochemical sense; all icebergs will likely fall somewhere between the basal and Monte Carlo approaches, but defining a ‘real’ scenario is presently impossible with information on the thickness, distribution and orientation of sediment layers in ice (see above comment).

Fig.2, l.164: “Days elapsed”/“residence time”: Iceberg drift is not explicitly modelled, as far as I understand, so that model icebergs will never leave the fjord? So the x-axis shows rather a potential “residence time” in the fjord and just refers to the “length of the model run” in days?
Yes, this is a more precise label. Ammended.

l. 163, l. 205: Delete “endmember”
Changed.

References referred to in all reviews:

1. Person, R. *et al.* Sensitivity of ocean biogeochemistry to the iron supply from the Antarctic ice sheet explored with a biogeochemical model. *Biogeosciences Discuss.* **2019**, 1–37 (2019).
2. Laufkötter, C., Stern, A. A., John, J. G., Stock, C. A. & Dunne, J. P. Glacial Iron Sources Stimulate the Southern Ocean Carbon Cycle. *Geophys. Res. Lett.* **45**, 13,377–13,385 (2018).
3. Shaw, T. J. *et al.* Input, composition, and potential impact of terrigenous material from free-drifting icebergs in the Weddell Sea. *Deep. Res. Part II-Topical Stud. Oceanogr.* **58**, 1376–1383 (2011).
4. Mugford, R. I. & Dowdeswell, J. A. Modeling iceberg-rafted sedimentation in high-latitude fjord environments. *J. Geophys. Res. Earth Surf.* **115**, (2010).
5. Warren, S. G., Roesler, C. S., Brandt, R. E. & Curran, M. Green Icebergs Revisited. *J. Geophys. Res. Ocean.* **124**, 925–938 (2019).
6. Herraiz-Borreguero, L., Lannuzel, D., van der Merwe, P., Treverrow, A. & Pedro, J. B. Large flux of iron from the Amery Ice Shelf marine ice to Prydz Bay, East Antarctica. *J. Geophys. Res. Ocean.* **121**, 6009–6020 (2016).
7. Duprat, L. P. A. M., Bigg, G. R. & Wilton, D. J. Enhanced Southern Ocean marine productivity due to fertilization by giant icebergs. *Nat. Geosci.* **9**, 219–221 (2016).
8. Wu, S.-Y. & Hou, S. Impact of icebergs on net primary productivity in the Southern Ocean. *Cryosph.* **11**, 707–722 (2017).
9. Schwarz, J. N. & Schodlok, M. P. Impact of drifting icebergs on surface phytoplankton biomass in the Southern Ocean: Ocean colour remote sensing and in situ iceberg tracking. *Deep. Res. Part I Oceanogr. Res. Pap.* **56**, 1727–1741 (2009).
10. Lin, H., Rauschenberg, S., Hexel, C. R., Shaw, T. J. & Twining, B. S. Free-drifting icebergs as sources of iron to the Weddell Sea. *Deep. Res. Part II-Topical Stud. Oceanogr.* **58**, 1392–1406 (2011).
11. Rogan, N. *et al.* Volcanic ash as an oceanic iron source and sink. *Geophys. Res. Lett.* **43**, 2732–2740 (2016).
12. Wuttig, K. *et al.* Impacts of dust deposition on dissolved trace metal concentrations (Mn, Al and Fe) during a mesocosm experiment. *Biogeosciences* **10**, 2583–2600 (2013).
13. Stephenson, G. R. *et al.* Subsurface melting of a free-floating Antarctic iceberg. *Deep Sea Res. Part II Top. Stud. Oceanogr.* **58**, 1336–1345 (2011).
14. Lin, H. & Twining, B. S. Chemical speciation of iron in Antarctic waters surrounding free-drifting icebergs. *Mar. Chem.* **128**, 81–91 (2012).
15. Boyd, P. W., Arrigo, K. R., Strzeppek, R. & Van Dijken, G. L. Mapping phytoplankton iron utilization: Insights into Southern Ocean supply mechanisms. *J. Geophys. Res. Ocean.* **117**, C06009 (2012).

16. Schroth, A. W., Crusius, J., Campbell, R. W. & Hoyer, I. Estuarine removal of glacial iron and implications for iron fluxes to the ocean. *Geophys. Res. Lett.* **41**, 3951–3958 (2014).
17. Zhang, R. *et al.* Transport and reaction of iron and iron stable isotopes in glacial meltwaters on Svalbard near Kongsfjorden: From rivers to estuary to ocean. *Earth Planet. Sci. Lett.* **424**, 201–211 (2015).
18. Maiti, K., Charette, M. A., Buesseler, K. O. & Kahru, M. An inverse relationship between production and export efficiency in the Southern Ocean. *Geophys. Res. Lett.* **40**, 1557–1561 (2013).
19. St-Laurent, P., Yager, P. L., Sherrell, R. M., Stammerjohn, S. E. & Dinniman, M. S. Pathways and supply of dissolved iron in the Amundsen Sea (Antarctica). *J. Geophys. Res. Ocean.* (2017). doi:10.1002/2017JC013162
20. St-Laurent, P. *et al.* Modeling the Seasonal Cycle of Iron and Carbon Fluxes in the Amundsen Sea Polynya, Antarctica. *J. Geophys. Res. Ocean.* **124**, 1544–1565 (2019).
21. Moon, T. *et al.* Subsurface iceberg melt key to Greenland fjord freshwater budget. *Nat. Geosci.* **11**, 49–54 (2018).
22. Hopwood, M. J. *et al.* Non-linear response of summertime marine productivity to increased meltwater discharge around Greenland. *Nat. Commun.* **9**, 3256 (2018).
23. Kanna, N. *et al.* Upwelling of macronutrients and dissolved inorganic carbon by a subglacial freshwater driven plume in Bowdoin Fjord, northwestern Greenland. *J. Geophys. Res. Biogeosciences* **123**, (2018).
24. Cape, M. R., Straneo, F., Beird, N., Bundy, R. M. & Charette, M. A. Nutrient release to oceans from buoyancy-driven upwelling at Greenland tidewater glaciers. *Nat. Geosci.* **12**, 34–39 (2019).
25. Hopwood, M. J. *et al.* Review Article: How does glacier discharge affect marine biogeochemistry and primary production in the Arctic? *Cryosph. Discuss.* **2019**, 1–51 (2019).
26. Lippitt, S. M., Lohan, M. C. & Bruland, K. W. The distribution of reactive iron in northern Gulf of Alaska coastal waters. *Mar. Chem.* **121**, 187–199 (2010).
27. Thuroczy, C.-E. *et al.* Key role of organic complexation of iron in sustaining phytoplankton blooms in the Pine Island and Amundsen Polynyas (Southern Ocean). *Deep. Res. Part II-Topical Stud. Oceanogr.* **71–76**, 49–60 (2012).
28. Sutherland, D. A. *et al.* Direct observations of submarine melt and subsurface geometry at a tidewater glacier. *Science (80-.)*. **365**, 369 LP – 374 (2019).
29. FitzMaurice, A. C., Cenedese, C. & Straneo, F. Nonlinear response of iceberg side melting to ocean currents. *Geophys. Res. Lett.* **44**, 5637–5644 (2017).
30. Yankovsky, A. E. & Yashayaev, I. Surface buoyant plumes from melting icebergs in the Labrador Sea. *Deep Sea Res. Part I Oceanogr. Res. Pap.* **91**, 1–9 (2014).
31. Shaw, T. J. *et al.* 234Th-Based Carbon Export around Free-Drifting Icebergs in the Southern Ocean. *Deep Sea Res. Part II Top. Stud. Oceanogr.* **58**, 1384–1391 (2011).
32. Smith, K. L. *et al.* Carbon export associated with free-drifting icebergs in the Southern Ocean. *Deep. Res. Part II Top. Stud. Oceanogr.* **58**, 1485–1496 (2011).
33. Vernet, M., Sines, K., Chakos, D., Cefarelli, A. O. & Ekern, L. Impacts on phytoplankton dynamics by free-drifting icebergs in the NW Weddell Sea. *Deep Sea Res. Part II Top. Stud. Oceanogr.* **58**, 1422–1435 (2011).
34. Henson, S., Le Moigne, F. & Giering, S. Drivers of Carbon Export Efficiency in the Global Ocean. *Global Biogeochem. Cycles* **33**, 891–903 (2019).
35. Smith, K. L., Sherman, A. D., Shaw, T. J. & Sprintall, J. Icebergs as Unique Lagrangian Ecosystems in Polar Seas. *Ann. Rev. Mar. Sci.* **5**, 269–287 (2013).
36. Klunder, M. B., Laan, P., Middag, R., De Baar, H. J. W. & Van Ooijen, J. C. Dissolved iron in the Southern Ocean (Atlantic sector). *Deep. Res. Part II* **58**, 2678–2694 (2011).

Reviewers' comments:

Reviewer #1 (Remarks to the Author):

The revised manuscript by Hopwood and co-authors is a welcome improvement on the original submission and I commend the authors on their changes and detailed response to reviewer comments. I still have some issues with the tone of the paper in places, and I don't necessarily agree with some of the responses, but I think this boils down mostly to difference of opinion and should not impede the publication of the manuscript. My suggestions are relatively minor, but I think should be addressed before publication.

Lines 33-35: I think a better way to phrase this would be something like "Future marine productivity may be not only sensitive to increasingly total icebergs fluxes, but also to changing iceberg properties, internal sediment distribution and melt dynamics."

Lines 63-65: Do many papers really suggest this flux is negligible? I don't think that seems to be a common assumption in the literature, especially given the studies mentioned.

Line 74-75: I think it's important to provide some brief information on what "total dissolvable Fe" actually is here i.e. pH ~2 leach for ~6 months.

Line 83: What size fraction (or particulate fraction?) is this value from? This seems to be filterable Fe, not a comparable acid leachable. Either way a short comment here would clear up any confusion.

Line 109: Reference needed

Line 123: I think a recent publication by Raiswell et al. (2018) could be cited here as well.

Line 148: I would argue the authors use information that is quite selective. If they included particulate Fe data here (either wet deposition of dust, or suspended particulate matter in rivers), then the values would span a much greater range – maybe not quite as great as iceberg Fe, but still much larger than indicated. In the context of this study (data from an acidic leach comprised most of the particulate fraction) this is important. If direct comparison are going to be made, then the "dissolved" Fe range from the limited number of icebergs sampled is within three orders of magnitude.

Fe release from melting icebergs: A useful paper recently published might be useful for contextual information in this section – Winter et al. (2019).

Line 284-288: I do not fully agree with the response to reviewer comments on this. The range in Fig 6d of this paper is somewhere between 7 and 300 $\mu\text{mol m}^{-2} \text{yr}^{-1}$ where icebergs would commonly be assumed to track. I don't think using a regionalised average makes sense (at least in my mind) because iceberg inputs are essentially limited to iceberg tracks and it would be unlikely that fertilization occurs much beyond these (for one "bioavailable" Fe would be utilized relatively rapidly in Fe starved regions). The point I was trying to make in my first review was not necessarily that the authors needed to use exact iceberg tracks, but that the authors should be clear in the uncertainties presented, as reviewer 2 also pointed out. Assuming a low Fe utilisation rate in a region that has areas with little to no iceberg input (albeit with regions of high input) and then directly comparing it to the "theoretical" supply based on concentrations isn't necessarily robust. I think the revised manuscript is clearer, but this direct comparison is potentially misleading without additional clarity.

Lines 302-303: See also previously mentioned paper by Raiswell et al. (2018) as I believe they provide other "dissolved" Fe measurements for icebergs.

Line 322: Check units here

Uncertainties in C export from iceberg fertilization: I think this section makes a valuable general point that high primary productivity does not necessarily lead to efficient carbon export, which is not very well represented in the models. However, (and this is more of a stylistic comment) which is personal preference, but I think this section feels bloated to make this point. Essentially the authors are highlighting that to fully understand Fe fertilization you need to understand C export efficiency. I don't think this is necessarily just a problem for iceberg inputs (and therefore this paper), and there is arguably a more general argument to be made for all Fe input mechanisms. This section could be trimmed to highlight this issue, and the uncertainties we need to overcome when address the role of icebergs in ocean Fe fertilization.

References:

Raiswell, R., Hawkings, J., Elsenousy, A., Death, R., Tranter, M., & Wadham, J. (2018). Iron in Glacial Systems: Speciation, Reactivity, Freezing Behavior, and Alteration During Transport. *Frontiers in Earth Science*, 6(222). <https://doi.org/10.3389/feart.2018.00222>

Winter, K., Woodward, J., Ross, N., Dunning, S. A., Hein, A. S., Westoby, M. J., ... Siegert, M. J. (2019). Radar-Detected Englacial Debris in the West Antarctic Ice Sheet. *Geophysical Research Letters*, 0(0). <https://doi.org/10.1029/2019GL084012>

Reviewer #2 (Remarks to the Author):

The manuscript, a resubmission by Hopwood et al., details, using a combination of field work, modeling efforts, and scaling arguments, the role of icebergs as iron sources to the ocean and their potential contribution to marine biogeochemical cycling. As pointed out for the previous submission, the authors analyze a unique observational dataset, which in concert with modeling output provides intriguing insight into iceberg-ocean interactions and their ramifications.

The authors have significantly edited the manuscript so as to incorporate reviewer feedback, presenting a more well-rounded and balanced discussion of results which nevertheless pushes the state of knowledge. While the observational dataset itself vastly increases our understanding of iron concentrations in icebergs, the modeling framework allows for testing of particular scenarios concerning iceberg iron contributions to the ocean, a novel approach which may contribute to future incorporation of iceberg fluxes in regional or global models.

Data and code necessary to reproduce results are made available by the authors, with the methods providing a clear description of sample collection and analysis steps.

Edits to the original manuscript, as well as responses to reviewer, have addressed my major concerns. The article is well-written, convincing, and deservant of prompt publication.

Reviewer #3 (Remarks to the Author):

In my 1st review, I highlighted how rare and difficult it is to obtain this dataset. And, as pointed in this paper, interpretations emanating from published papers are contradict each other. This work is timely as the community is trying to understand the effects of increased ice discharge on polar ecosystems, and increased international efforts now focus on ice shelves (e.g. Thwaites Program), with an emphasis on long term monitoring.

I have read the comments from the other reviewers. The answers from the authors are well researched and convincing.

The paper is extremely well thought, with a clear hypothesis, and a combination of observations and modelling approaches to address it.

The authors may want to refer to the recently published work by Person et al. 2019 (Sensitivity of ocean biogeochemistry to the iron supply from the Antarctic Ice Sheet explored with a biogeochemical model) in their manuscript.

Reviewer #4 (Remarks to the Author):

The authors have performed very solid work and I am overall very pleased with the response to my review.

In particular, the amended Figure 1 is now much better understandable and the concern of regional dependence of the plot got addressed by adding more distributions to panel (b).

With the additional supplement it is now also much clearer that a substantial amount of the collected data was newly introduced in this study.

I also acknowledge the further additions to the paper concerning the discussion of the different melting cases. The iceberg model formulation also improved, although some of the mentioned leading papers of the field could have been cited here in my view. But I will leave this up to the authors.

As a last comment, the structural uncertainty of the iceberg model concerning the wave erosion parameterization was also addressed, so that I can suggest the paper to be published in its current form.

Thomas Rackow

Reviewer #1 (Remarks to the Author):

The revised manuscript by Hopwood and co-authors is a welcome improvement on the original submission and I commend the authors on their changes and detailed response to reviewer comments. I still have some issues with the tone of the paper in places, and I don't necessarily agree with some of the responses, but I think this boils down mostly to difference of opinion and should not impede the publication of the manuscript. My suggestions are relatively minor, but I think should be addressed before publication.

Lines 33-35: I think a better way to phrase this would be something like "Future marine productivity may be not only sensitive to increasingly total icebergs fluxes, but also to changing iceberg properties, internal sediment distribution and melt dynamics."

Reply: Yes this is more precise and changed accordingly.

Lines 63-65: Do many papers really suggest this flux is negligible? I don't think that seems to be a common assumption in the literature, especially given the studies mentioned.

Reply: In studies primarily concerning icebergs this is obviously not the case, but there is an inherent bias in any 'iceberg-themed' paper reference list (including in this manuscript) towards literature that does include an extensive discussion of iceberg Fe fluxes. If we look more broadly, for example at the global scale using Fe models, most of the 13 state of the art Fe-models compared by Tagliabue et al.,¹ contain neither a specific glacier or iceberg Fe source. To someone who works in the field of glacier/iceberg biogeochemistry that might seem odd, but if we look at models in more detail (I will comment only on MEDUSA because it is the only one I am familiar with), MEDUSA does a fantastic job of reproducing the very dynamic observed distribution of dissolved Fe around Greenland, it captures the strong gradient in Fe concentrations observed across the eastern coastline transition to the N Atlantic, it captures Fe-limitation in the Irminger Basin and shows a relatively good fit to all measured biogeochemical data. But it includes no specific glacier/iceberg source; freshwater and the terrestrial/shelf/coastal influence is parameterized as per the rest of the 'unglaciated' N hemisphere as a function of rain deposition and sediment surface area. So on the one hand some glaciologists/iceberg specialists often argue something is missing in this type of model (in bulk terms, using data from recent Raiswell et al., work icebergs should be the dominant Fe input into near-shore waters around Greenland). But from an observational perspective the model does a near-perfect job without any glacier/iceberg associated processes exerting any influence at all on this scale. The reason why no further parametrization for ice-ocean interaction has been included is because it's not thought to be important on this scale i.e. any additions and subsequent Fe removal are occurring 'sub-scale' i.e. either within the fjords around the island or on scales much smaller than the model grid in the open ocean as isolated Fe 'hotspots'. So with respect to broad-scale Fe availability in the ocean, we think it is correct to state that there is a general assumption that the net effect of icebergs is negligible at large scales.

Line 74-75: I think it's important to provide some brief information on what "total dissolvable Fe" actually is here i.e. pH ~2 leach for ~6 months.

Reply: yes this would read better. Amended to read: 'Total dissolvable Fe (hereafter, 'Fe'), Fe which is soluble in weak HCl after 6 months at pH <2,'

Line 83: What size fraction (or particulate fraction?) is this value from? This seems to be filterable Fe, not a comparable acid leachable. Either way a short comment here would clear up any confusion.

Reply: no it's TdFe, amended for clarity 'the median TdFe concentration'

Line 109: Reference needed

Reply: Literature on atmospheric dust uses a broad range of definitions for 'coarse' and 'fine'. Most commonly 1-10, 2-10, 2.5-10 or 1/2/3+ μm defines 'coarse'. There is apparently no standard definition in oceanographic literature and we can find papers with each of these-and other, or no- ranges quoted, but '2.5-10 μm ' is the official designation by the US EPA so we can define this in the text '(~3-10 μm , US Environmental Protection Agency definition')

Line 123: I think a recent publication by Raiswell et al. (2018) could be cited here as well.

We have referred to this reference earlier in the text, but don't use it here for a specific reason. The study² in theory 'updates' many of the values we quote from the 2016 study by similar authors³ on a related topic, in addition to more general discussion of the cycling of Fe in the cryosphere. We stuck with the 2016 values concerning mean iceberg Fe content (defined from labile sediment) for simplicity because it is not clear how to 'weight' data from the 2018 study: In the 2016 paper a simple global mean labile sediment content value is given, which combined with a mean sediment loading (0.5 g per L of ice) gives a simple global average iceberg labile Fe content of 6.8 μM . It is clear that the purpose of this section of the 2016 paper was to define such a mean and it's broadly useful for comparative purposes. It's less clear how to treat the 2018 'update' as it is more of a process study discussion.

If we took the 'new' values in the 2018 paper for Greenland which have a higher %wt. labile Fe than the 2016 values, we can deduce a higher Fe iceberg endmember of 11 μM . However, Raiswell 2018 also merges their existing data (2016) with our data from Svalbard (which is collected by melting larger samples of ice rather than chiseling out the sediment close to source)⁴. Our sediment weighted values are, on average, lower- and combined they produce a mean of 3 μM . We

commented previously on these values and suggest that the reason why they are low is because of the loss of basal ice as icebergs move offshore. If sediment-rich layers are preferentially lost from ice (as outlined in this paper), and sediment-rich layers contain the most labile Fe by %wt (which it's thought they do), then the 'labile Fe' content of particles remaining in an iceberg should, on average, decline as icebergs move offshore-potentially rapidly so in a warm Arctic catchment with high basal sediment loads (Kongsfjorden meets both of these criteria). This potentially invalidates any direct comparison between samples collected from sediment-rich layers near-shore and melted iceberg samples a little further offshore. We cannot therefore see any easy way of using the values presented in Raiswell 2018 to meaningfully update the 2016 'global mean' and for this reason opted to stick with the value from the slightly earlier paper.

Line 148: I would argue the authors use information that is quite selective. If they included particulate Fe data here (either wet deposition of dust, or suspended particulate matter in rivers), then the values would span a much great range – maybe not quite as great as iceberg Fe, but still much larger than indicated. In the context of this study (data from an acidic leach comprised most of the particulate fraction) this is important. If direct comparison are going to be made, then the “dissolved” Fe range from the limited number of icebergs sampled is within three orders of magnitude.

Reply: The reviewer is correct that this is not a particularly detailed comparison. We can expand this a little and it is worth considering both dissolved and total composition. We should also note the difference between single site, and global datasets. Comparison to dust is not straightforward as dust deposition at the sea surface is not directly measured. The easiest and most valid comparisons are other freshwater sources where [concentration × volume] approximates the flux i.e. rainwater and river water. In constructing the below table we have tried to find the largest/most appropriate data compilations. It is apparent that the limited iceberg and glacier 'total' concentrations exhibit a very broad range compared to other freshwater sources. The comparison is however flawed in one sense that there are many more riverine studies available and these datasets tend to be far larger. The detection limits and method techniques (especially definition of 'total' Fe) also vary in the studies below. To avoid a lengthy discussion of these factors we have amended the sentence to read, 'with the range of measured concentrations being large compared to other freshwater-associated Fe supply mechanisms. For example, total Fe concentrations in rainwater range over two orders of magnitude from ~10–1000 nM^{5,6}, and river water from ~0.1–100 μM^{7,8}.'

	Dissolved (nM)	Total dissolvable or total labile (nM)
Rain	88-300 ⁹	17-1200* (Northern Hemisphere ⁶) 7-100* (Southern Hemisphere ⁶) 13-1400 (North Carolina ⁶)
Rivers	<detection-19000* ⁸ 1100-8100 ¹⁰ 870-12000 ¹¹	<detection-228000* ⁸ 2100-38000 ¹⁰ 720-580000 ¹²
Icebergs	4-610 ¹³ 1-540 ⁴ <detection-300 ¹⁴	2-4200 (This study, Godthabsfjord) 26-57000 (This study, Kongsfjorden) 6-150000 (This study, South Bay) 2-1900000*
Glacier fed streams	230-4700 ¹⁵ 18-180 ¹⁶	17000-100000 ¹⁵

*Indicates a global or broad regional data compilation. As many river studies are available we have selected the largest we could find with the broadest range of Fe. Large datasets for glaciers and rain are sparse by comparison.

Fe release from melting icebergs: A useful paper recently published might be useful for contextual information in this section – Winter et al. (2019).

Reply: Yes this new reference is added to the line : 'This may especially be the case for ice calved from large ice shelves compared to ice calved from inshore marine-terminating glaciers, as the pathways for sediment incorporation into ice (and thus sediment loss from ice) differ between these scenarios and are not well characterized.'

Line 284-288: I do not fully agree with the response to reviewer comments on this. The range in Fig 6d of this paper is somewhere between 7 and 300 umol m-2 yr-1 where icebergs would commonly be assumed to track. I don't think using a regionalised average makes sense (at least in my mind) because iceberg inputs are essentially limited to iceberg tracks and it would be unlikely that fertilization occurs much beyond these (for one “bioavailable” Fe would be utilized relatively rapidly in Fe starved regions). The point I was trying to make in my first review was not necessarily that the authors needed to use exact iceberg tracks, but that the authors should be clear in the uncertainties presented, as reviewer 2 also pointed out. Assuming a low Fe utilisation rate in a region that has areas with little to no iceberg input (albeit with regions of high input) and then directly comparing it to the “theoretical” supply based on concentrations isn't necessarily robust. I think the revised manuscript is clearer, but this direct comparison is potentially misleading without additional clarity.

Reply: Thank you for the clarification. This is the essential the point we follow up in Table 1 and Figure 3. If iceberg Fe is confined to 'small' regions, the maximum plausible primary production that can be simulated is much less than in a hypothetical calculation where the total Fe input is scaled to C export with no consideration of the area over which this

effect operates (because of the capping effect of ligands, and the maximum possible primary production that can be stimulated per unit area). For clarity we raise the issue of scale at the end of this section: ‘Furthermore, it is unclear on how broad a scale Fe-fertilization from icebergs operates. An influence of iceberg passage is generally observed on phytoplankton and nutrient distributions in iceberg tracks during the growth season, but the regional scale effects of icebergs are more challenging to deduce from satellite derived data alone and are still subject to considerable uncertainties between models. Tracking areas of iceberg Fe-enrichment is inherently challenging and thus any calculations to establish utilization rates are more challenging than for some other Fe sources.’

Lines 302-303: See also previously mentioned paper by Raiswell et al. (2018) as I believe they provide other “dissolved” Fe measurements for icebergs.

Reply: The Raiswell et al., 2018 manuscript uses slightly different terminology to define Fe phases, as far as we can tell it does not contain newly published dissolved Fe data, that is not included in the 2016 manuscript or in the sources already referred to herein.

Line 322: Check units here/ The units are correct.

Uncertainties in C export from iceberg fertilization: I think this section makes a valuable general point that high primary productivity does not necessarily lead to efficient carbon export, which is not very well represented in the models. However, (and this is more of a stylistic comment) which is personal preference, but I think this section feels bloated to make this point. Essentially the authors are highlighting that to fully understand Fe fertilization you need to understand C export efficiency. I don’t think this is necessarily just a problem for iceberg inputs (and therefore this paper), and there is arguably a more general argument to be made for all Fe input mechanisms. This section could be trimmed to highlight this issue, and the uncertainties we need to overcome when address the role of icebergs in ocean Fe fertilization.

Reply: This point is obviously not well understood as it’s not mentioned at all in many iceberg model studies or studies converting between Fe and C, and was subject to many questions by reviewers (and indeed several authors when we began working on the dataset). Yes it essentially affects all Fe sources, but is particularly important for icebergs because of the uncertainty in the spatial scale of this specific fertilization effect. As raised by the reviewer, there is a massive difference between a scenario where iceberg fertilization is confined to narrow iceberg tracks, and one where icebergs have a broad-scale fertilizing effect. The primary production, and primary production-C export relationships between these two scenarios likely also vary, and this difference is one key reason why estimates of iceberg-Fe-fertilization vary so much in current literature. For several other Fe-sources (e.g. dust) the relationship between Fe input and C export is less problematic because there isn’t such a difficulty in constraining the scale over which the Fe source operates (i.e. there are large areas of the ocean where dust is, to a first-order approximation, the only major Fe source).

Reviewer #2 (Remarks to the Author):

The manuscript, a resubmission by Hopwood et al., details, using a combination of field work, modeling efforts, and scaling arguments, the role of icebergs as iron sources to the ocean and their potential contribution to marine biogeochemical cycling. As pointed out for the previous submission, the authors analyze a unique observational dataset, which in concert with modeling output provides intriguing insight into iceberg-ocean interactions and their ramifications. The authors have significantly edited the manuscript so as to incorporate reviewer feedback, presenting a more well-rounded and balanced discussion of results which nevertheless pushes the state of knowledge. While the observational dataset itself vastly increases our understanding of iron concentrations in icebergs, the modeling framework allows for testing of particular scenarios concerning iceberg iron contributions to the ocean, a novel approach which may contribute to future incorporation of iceberg fluxes in regional or global models. Data and code necessary to reproduce results are made available by the authors, with the methods providing a clear description of sample collection and analysis steps. Edits to the original manuscript, as well as responses to reviewer, have addressed my major concerns. The article is well-written, convincing, and deserving of prompt publication.

Reply: The reviewer is thanked for their final comments on the manuscript.

Reviewer #3 (Remarks to the Author):

In my 1st review, I highlighted how rare and difficult it is to obtain this dataset. And, as pointed in this paper, interpretations emanating from published papers are contradict each other. This work is timely as the community is trying to understand the effects of increased ice discharge on polar ecosystems, and increased international efforts now focus on ice shelves (e.g. Thwaites Program), with an emphasis on long term monitoring. I have read the comments from the other reviewers. The answers from the authors are well researched and convincing. The paper is extremely well thought, with a clear hypothesis, and a combination of observations and modelling approaches to address it. The authors may want to refer to the recently published

work by Person et al. 2019 (Sensitivity of ocean biogeochemistry to the iron supply from the Antarctic Ice Sheet explored with a biogeochemical model) in their manuscript.

Reply: The reviewer is thanked for their final comments on the manuscript. The Person et al., text we had already noted and included in the original submission as a ‘biogeosciences discussions’ paper which seemed robust and relevant. We have updated the reference since the paper is now published, as far as we can tell there are no major changes that affect the discussion of the Person et al., text herein.

Reviewer #4 (Remarks to the Author):

The authors have performed very solid work and I am overall very pleased with the response to my review. In particular, the amended Figure 1 is now much better understandable and the concern of regional dependence of the plot got addressed by adding more distributions to panel (b). With the additional supplement it is now also much clearer that a substantial amount of the collected data was newly introduced in this study. I also acknowledge the further additions to the paper concerning the discussion of the different melting cases. The iceberg model formulation also improved, although some of the mentioned leading papers of the field could have been cited here in my view. But I will leave this up to the authors. As a last comment, the structural uncertainty of the iceberg model concerning the wave erosion parameterization was also addressed, so that I can suggest the paper to be published in its current form.

Reply: The reviewer is thanked for their final comments on the manuscript. Our submission was slightly over the recommended references but, unless there are editorial objections, we can add a few key references that may be of use to the reader when raising the issue of wave-melt parameterization (now added):

Bigg, G. R., Wadley, M. R., Stevens, D. P. & Johnson, J. A. Modelling the dynamics and thermodynamics of icebergs. *Cold Reg. Sci. Technol.* **26**, 113–135 (1997).

Silva, T. A. M., Bigg, G. R. & Nicholls, K. W. Contribution of giant icebergs to the Southern Ocean freshwater flux. *J. Geophys. Res. Ocean.* **111**, (2006).

References referred to:

1. Tagliabue, A. *et al.* How well do global ocean biogeochemistry models simulate dissolved iron distributions? *Global Biogeochem. Cycles* **30**, 149–174 (2016).
2. Raiswell, R. *et al.* Iron in Glacial Systems: Speciation, Reactivity, Freezing Behavior, and Alteration During Transport. *Front. Earth Sci.* **6**, 1–17 (2018).
3. Raiswell, R. *et al.* Potentially bioavailable iron delivery by iceberg-hosted sediments and atmospheric dust to the polar oceans. *Biogeosciences* **13**, 3887–3900 (2016).
4. Hopwood, M. J., Cantoni, C., Clarke, J. S., Cozzi, S. & Achterberg, E. P. The heterogeneous nature of Fe delivery from melting icebergs. *Geochemical Perspect. Lett.* **3**, 200–209 (2017).
5. Kieber, R. J., Willey, J. D. & Avery, G. B. Temporal variability of rainwater iron speciation at the Bermuda Atlantic time series station. *J. Geophys. Res.* **108**, (2003).
6. Kieber, R. J., Williams, K., Willey, J. D., Skrabal, S. & Avery, G. B. Iron speciation in coastal rainwater: concentration and deposition to seawater. *Mar. Chem.* **73**, 83–95 (2001).
7. Neal, C. *et al.* Increasing iron concentrations in UK upland waters. *Aquat. Geochemistry* **14**, 263–288 (2008).
8. Neal, C. & Robson, A. J. A summary of river water quality data collected within the Land-Ocean Interaction Study: core data for eastern UK rivers draining to the North Sea. *Sci. Total Environ.* **251**, 585–665 (2000).
9. Zhuang, G., Yi, Z. & Wallace, G. T. Iron(II) in rainwater, snow, and surface seawater from a coastal environment. *Mar. Chem.* **50**, 41–50 (1995).
10. Ponter, C., Ingri, J., Burman, J. O. & Bostrom, K. TEMPORAL VARIATIONS IN DISSOLVED AND SUSPENDED IRON AND MANGANESE IN THE KALIX RIVER, NORTHERN SWEDEN. *Chem. Geol.* **81**, 121–131 (1990).
11. Lofts, S., Tipping, E. & Hamilton-Taylor, J. The Chemical Speciation of Fe(III) in Freshwaters. *Aquat. Geochemistry* **14**, 337–358 (2008).
12. Sahoo, P. K. *et al.* High resolution hydrogeochemical survey and estimation of baseline concentrations of trace elements in surface water of the Itacaiúnas River Basin, southeastern Amazonia: Implication for environmental studies. *J. Geochemical Explor.* **205**, 106321 (2019).
13. Lin, H. & Twining, B. S. Chemical speciation of iron in Antarctic waters surrounding free-drifting icebergs. *Mar. Chem.* **128**, 81–91 (2012).
14. Hopwood, M. J. *et al.* Seasonal changes in Fe along a glaciated Greenlandic fjord. *Front. Earth Sci.* **4**, (2016).
15. Hawkings, J. R. *et al.* Ice sheets as a significant source of highly reactive nanoparticulate iron to the oceans. *Nat. Commun.* **5**, (2014).
16. Statham, P. J., Skidmore, M. & Tranter, M. Inputs of glacially derived dissolved and colloidal iron to the coastal ocean and implications for primary productivity. *Global Biogeochem. Cycles* **22**, (2008).

REVIEWERS' COMMENTS:

Reviewer #1 (Remarks to the Author):

I am happy with responses and the changes the authors have made to the paper. The manuscript is suitable for publication in its current form.